# Provable benefits of annealing for estimating normalizing constants: Importance Sampling, Noise-Contrastive Estimation, and beyond

**Omar Chehab**
Université Paris-Saclay, Inria, CEA
Palaiseau, France
`l-emir-omar.chehab@inria.fr`

**Aapo Hyvärinen**
Department of Computer Science
University of Helsinki
Helsinki, Finland
`aapo.hyvarinen@helsinki.fi`

**Andrej Risteski**
Department of Machine Learning
Carnegie-Mellon University
Pittsburgh, USA
`aristesk@andrew.cmu.edu`

## Abstract

Recent research has developed several Monte Carlo methods for estimating the normalization constant (partition function) based on the idea of annealing. This means sampling successively from a path of distributions that interpolate between a tractable "proposal" distribution and the unnormalized "target" distribution. Prominent estimators in this family include annealed importance sampling and annealed noise-contrastive estimation (NCE). Such methods hinge on a number of design choices: which estimator to use, which path of distributions to use and whether to use a path at all; so far, there is no definitive theory on which choices are efficient. Here, we evaluate each design choice by the asymptotic estimation error it produces. First, we show that using NCE is more efficient than the importance sampling estimator, but in the limit of infinitesimal path steps, the difference vanishes. Second, we find that using the geometric path brings down the estimation error from an exponential to a polynomial function of the parameter distance between the target and proposal distributions. Third, we find that the arithmetic path, while rarely used, can offer optimality properties over the universally-used geometric path. In fact, in a particular limit, the optimal path is arithmetic. Based on this theory, we finally propose a two-step estimator to approximate the optimal path in an efficient way.

## 1 Introduction

Recent progress in generative modeling has sparked renewed interest in models of data that are defined by an unnormalized distribution. A prominent example is energy-based models, which are increasingly used in deep learning [1], and for which there are a variety of parameter estimation procedures [2–5]. Another example comes from Bayesian statistics, where the posterior model of parameters given data is frequently known only up to a proportionality constant. Such models can be evaluated and compared by the probability they assign to a dataset, yet this requires computing their normalization constants (partition functions) which are typically high-dimensional, intractable integrals.

37th Conference on Neural Information Processing Systems (NeurIPS 2023).

Monte-Carlo techniques have been successful at computing these integrals using sampling methods [6]. The most common is importance sampling [6] which draws a sample from a tractable, "proposal" distribution to integrate the unnormalized "target" density. Noise-contrastive estimation (NCE) [3] uses a sample from *both* the proposal and the target, to compute the integral. Yet such methods suffer from high variance, especially when the "gap" between the proposal and target densities is large [7–9]. This has motivated various approaches to gradually bridge the gap with intermediate distributions, which is loosely referred to as "annealing". Among them, annealed importance sampling (AIS) [10–12] is widely adopted: it has been used to compute the normalization constants of deep stochastic models [13, 14] or to motivate a lower-bound for learning objectives [15, 16]. To integrate the unnormalized "target" density, it draws a sample from an entire path of distributions between the proposal and the target. While annealed importance sampling has been shown to be effective empirically, its theoretical understanding remains limited [17, 18]: it is yet unclear when annealing is effective, for which annealing paths, and whether AIS is a statistically efficient way to do it.

In this paper, we define a family of *annealed Bregman estimators (ABE)* for the normalization constant. We show that it is general enough to recover many classical estimators as a special case, including importance sampling, noise-contrastive estimation, umbrella sampling [19], bridge sampling [20] and annealed importance sampling. We provide a statistical analysis of its hyperparameters such as the choice of paths, and show the following:

1. First, we establish that using NCE is more asymptotically statistically efficient — in the sense of how many samples from the intermediate distribution need to be generated — than the importance sampling estimator, but in the limit of infinitesimal path steps, the difference vanishes.

2. Second, we find that the near-universally used *geometric path* brings down the estimation error from an exponential to a polynomial function of the parameter distance between the target and proposal distributions.

3. Third, we find that using the recently introduced *arithmetic path* [21] is exponentially inefficient in its basic form, yet it can be reparameterized to be in some sense optimal. Based on this optimality result, we finally propose a two-stage estimation procedure which first finds an approximation of the optimal (arithmetic) path, then uses it to estimate the normalization constant.

## 2 Background

**Importance sampling and NCE**  The problem considered here is computing the normalization constant [1], *i.e.* the integral of some unnormalized density $f_1(\boldsymbol{x})$ called "target".

Importance sampling and noise-contrastive estimation are two common estimators for that integral which use a random sample drawn from a tractable density $p_0(\boldsymbol{x})$ called "proposal"(Table 1, column 3). In fact, they are part of a larger family of Monte-Carlo estimators which can be interpreted as solving a binary classification task, aiming to distinguish between a sample drawn from the proposal and another from the target [22]. These estimators are summarized in Table 1. Each estimator is obtained by minimizing a specific binary classification loss that is identified by a convex function $\phi(\boldsymbol{x})$. For example, minimizing the classification loss identified by $\phi_{\text{IS}}(x) = x \log x$ yields the importance sampling estimator. Similarly, $\phi_{\text{RevIS}}(x) = -\log x$ leads to the reverse importance sampling estimator [23], and $\phi_{\text{NCE}}(x) = x \log x - (1 + x) \log((1 + x)/2)$ to the noise-contrastive estimator.

**Annealed estimators**  Annealing extends the above "binary" setup, by introducing a sequence of $K + 1$ distributions from the proposal to the target (included). The idea will be to draw a sample from *all* these distributions to estimate the integral of the target $f_1(\boldsymbol{x})$.

These intermediate distributions are obtained from a path $(f_t)_{t \in [0,1]}$, defined by interpolating between the proposal $p_0$ and unnormalized target $f_1$: this path is therefore unnormalized. Different interpolation schemes can be chosen. A general one, explained in Figure 1, is to take the $q$-mean of the

---

[1] in this paper we also say we "estimate" the normalization constant, though in classical statistics it is more traditional to use "estimation" when referring to parameters of a statistical model

| Name | Loss identified by $\phi(\boldsymbol{x})$ | Estimator $\hat{Z}_1$ | MSE |
|------|------|------|------|
| IS | $x \log x$ | $\mathbb{E}_{p_0} \frac{f_1}{p_0}$ | $\frac{1+\nu}{\nu N} \mathcal{D}_{\chi^2}(p_1, p_0)$ |
| RevIS | $-\log x$ | $\left( \mathbb{E}_{p_1} \frac{p_0}{f_1} \right)^{-1}$ | $\frac{1+\nu}{N} \mathcal{D}_{\chi^2}(p_0, p_1)$ |
| NCE | $x \log x - (1+x)\log(\frac{1+x}{2})$ | implicit | $\frac{(1+\nu)^2}{\nu N} \frac{\mathcal{D}_{\mathrm{HM}}(p_1, p_0)}{1 - \mathcal{D}_{\mathrm{HM}}(p_1, p_0)}$ |

Table 1: Some estimators of the normalization obtained by minimizing a classification loss, and their estimation error in terms of well-known divergences [22]. For details and definitions, see Appendix A.

| | |
|---|---|
| General ($q \in ]0,1]$) | $f_t(\boldsymbol{x}) = \left( (1-t)p_0(\boldsymbol{x})^q + t f_1(\boldsymbol{x})^q \right)^{\frac{1}{q}}$ |
| Geometric ($q \to 0$) | $f_t(\boldsymbol{x}) = p_0(\boldsymbol{x})^{1-t} \times f_1(\boldsymbol{x})^t$ |
| Arithmetic ($q = 1$) | $f_t(\boldsymbol{x}) = (1-t)p_0(\boldsymbol{x}) + t f_1(\boldsymbol{s})$ |

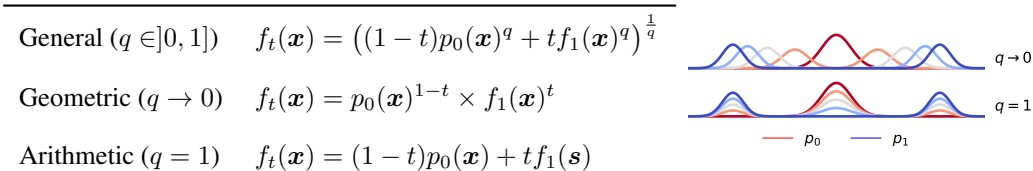

Figure 1: $q$-mean paths between the proposal and target distributions. The geometric and arithmetic paths are obtained as limit cases. Here, the proposal (red) is a standard Gaussian. The target (blue) is a Gaussian mixture with two modes, and same first and second moments as the proposal. The path of distributions interpolates between blue and red.

proposal and target [21]. Two values of $q$ are of particular interest: $q \to 0$ defines a a near-universal path [18], obtained by taking the geometric mean of the target and proposal, while $q = 1$ defines a path obtained by the arithmetic mean.

Once a path is chosen, it can be uniformly[2] discretized into a sequence of $K + 1$ unnormalized densities, denoted by $(f_{k/K})_{k \in [0,K]}$ with corresponding normalizations $(Z_{k/K})_{k \in [0,K]}$. In practice, samples are drawn from the corresponding normalized densities $(p_{k/K})_{k \in [0,K]}$ using Markov Chain Monte Carlo (MCMC). This sampling step incurs a computational cost, which is paid in the hope of reducing the variance of the estimation. It is common in the literature [17, 18] to assume *perfect sampling*, meaning the MCMC has converged and produced exact and independent samples from the distributions along the path, which simplifies the analysis.

**Estimation error**  A measure of "quality" is required to compare different estimation choices, such as whether to anneal and which path to use. Such a measure is given by the Mean Squared Error (MSE), which is generally tractable when written at the first order in the asymptotic limit of a large sample size [24, Eq. 5.20]. These expressions have been derived for estimators obtained by minimimizing a classification loss [22] and are included in Table 1. They measure the "gap" between the proposal and target distributions using statistical divergences. Note also that the estimation error depends on the *normalized* target density (column 4), while the estimators are computed using the *unnormalized* target density (column 3). Further details are available in Appendix A.

## 3   Annealed Bregman Estimators of the normalization constant

The question that we try to answer in this paper is: *How should we choose the $K$ distributions in annealing (the target is fixed), and how are their samples best used?* To answer this, we will study the error produced by different estimation choices. But first we define the set of estimators for which the analysis is done.

**Definition of Annealed Bregman Estimators**  We now define a new family of estimators, which we call *annealed Bregman estimators (ABE)*; the motivation for this terminology will become clear in the coming paragraphs. We will show that this is a general class of estimators for computing the normalization using a sample drawn from the sequence of $K + 1$ distributions. For ABE, the log

---

[2]other discretization schemes can be equivalently achieved by re-parameterizing the path [17]

normalization $\log Z_1$ is estimated additively along the sequence of distributions

$$\widehat{\log Z_1} = \sum_{k=0}^{K-1} \log\left(\widehat{\frac{Z_{(k+1)/K}}{Z_{k/K}}}\right) + \log Z_0 \ . \tag{1}$$

Defining the estimation in log-space is analytically convenient, as it is easier to analyze a sum of estimators than a product. Exponentiating the result leads to an estimator of $Z_1$. We naturally extend the binary setup ($K = 1$) of Chehab et al. [22] and propose to compute each of the intermediate log-ratios, by solving a classification task between samples drawn from their corresponding densities $p_{k/K}$ and $p_{(k+1)/K}$. Each binary classification loss is now identified by a convex function $\phi_k(\boldsymbol{x})$ and defined as

$$\mathcal{L}_k(\beta_k) := \mathbb{E}_{\boldsymbol{x}\sim p_{k/K}}[\phi_k^{'}(r_k(\boldsymbol{x};\beta_k)) \times r_k(\boldsymbol{x};\beta_k) - \phi_k(r_k(\boldsymbol{x};\beta_k))] - \mathbb{E}_{\boldsymbol{x}\sim p_{(k+1)/K}}[\phi_k^{'}(r_k(\boldsymbol{x};\beta_k))], \tag{2}$$

where the regression function $r_k(\boldsymbol{x};\beta_k)$ is parameterized by the unknown log-ratio $\beta_k$

$$r_k(\boldsymbol{x};\beta_k) = \exp(-\beta_k) \times f_{(k+1)/K}(\boldsymbol{x})/f_{k/K}(\boldsymbol{x}) \ . \tag{3}$$

For the true $\beta_k^*$, it holds $\beta_k^* = \log(Z_{(k+1)/K}/Z_{k/K})$ and $r_k(\boldsymbol{x};\boldsymbol{\beta}_k^*) = p_{(k+1)/K}(\boldsymbol{x})/p_{k/K}(\boldsymbol{x})$. The convex functions $(\phi_k)_{k\in[0,K-1]}$ which identify the classification losses are called "Bregman" generators, hence ABE. As mentioned above, we assume perfect sampling and allocate the total sample size $N$ equally among the $K$ estimators in the sum.

**Hyperparameters**  The annealed Bregman estimator depends on the following hyperparameters: (1) the choice of path $q$; (2) the number of distributions along that path $K + 1$ (including the proposal and the target); (3) the classification losses identified by the convex functions $(\phi_k)_{k\in[0,K-1]}$.

Different combinations of these hyperparameters recover several common estimators of the log-partition function. In binary case of $K = 1$ this includes importance sampling, reverse importance sampling, and noise-contrastive estimation, each obtained for a different choice of the classification loss [22]. To build intuition, consider $K = 2$ so that we add a single intermediate distribution $p_{1/2}$ to the sequence. Using the importance sampling loss ($\phi_0 = x \log x$) for the first ratio, and reverse importance sampling ($\phi_1 = -\log x$) for the second ratio, recovers the *bridge sampling estimator* as a special case [20]

$$\widehat{\log Z_1} = -\log \mathbb{E}_{p_1}\frac{f_{1/2}}{f_1} + \log \mathbb{E}_{p_0}\frac{f_{1/2}}{f_0} + \log Z_0 \ . \tag{4}$$

Alternatively, we can use these classification losses in reverse order: reverse importance sampling ($\phi_0 = -\log x$) for the first ratio, and importance sampling ($\phi_1 = x \log x$) for the second ratio, and recover the *umbrella sampling estimator* [19] also known as the *ratio sampling estimator* [25]

$$\widehat{\log Z_1} = \log \mathbb{E}_{p_{1/2}}\frac{f_1}{f_{1/2}} - \log \mathbb{E}_{p_{1/2}}\frac{f_0}{f_{1/2}} + \log Z_0 \ . \tag{5}$$

Another option yet, is to use the same classification loss for all ratios. With importance sampling ($\phi_k = x \log x, \forall k \in [\![0, K-1]\!]$), we recover the *annealed importance sampling* estimator [10–12]

$$\widehat{\log Z_1} = \sum_{k=1}^{K} \log \mathbb{E}_{\boldsymbol{x}\sim p_{k-1}}\left[\frac{f_k}{f_{k-1}}(\boldsymbol{x})\right] + \log Z_0 \ . \tag{6}$$

The family of annealed Bregman estimators is visibly large enough to include many existing estimators, obtained for different hyperparmeter choices. This raises the fundamental question of how these hyperparameters should be chosen, in particular in the challenging case where the *target and proposal have little overlap and the data is high dimensional*. To answer this question, we will study the estimation error produced by different hyperparameter choices.

## 4  Statistical analysis of the hyperparameters

We consider a fixed data budget $N$ and investigate how the remaining hyperparameters are best chosen for statistical efficiency. The starting point for the analysis is that as ABE estimates the

normalization in log-space, the estimator is obtained by a sum of independent and asymptotically unbiased estimators [26] given in Eq. 1 and thus the mean squared errors written in table 1 are additive. (Recall, the independence of these estimators is because new samples are drawn for each estimation task.) Each individual error actually measures an overlap between two consecutive distributions along the path, and annealing integrates these overlaps.

## 4.1 Classification losses, $\phi_k$

Given the popularity of annealed importance sampling, we should first ask if the importance sampling loss is really an acceptable default. We recall an important limitation of importance sampling: its estimation error is notoriously sensitive to distribution tails [27]. Without annealing, it is infinite when the target $p_1$ has a heavier tail than the proposal $p_0$. When annealing with a geometric path, for example between two Gaussians with different covariances $p_0 = \mathcal{N}(\mathbf{0}, \mathbf{Id})$ and $p_1 = \mathcal{N}(\mathbf{0}, 2\,\mathbf{Id})$, the geometric path produces Gaussians with increasing variances $\Sigma_t = (1 - t/2)^{-1}\,\mathbf{Id}$ and therefore increasing tails. Hence, the same tail mismatch holds along the path. Note that this concern is a realistic one for natural image data, as the target distribution over images is typically super-Gaussian [28] while the proposal is usually chosen as Gaussian.

This warrants a better choice for the loss: In the binary setup ($K = 1$), the NCE loss is optimal [20, 22] and its error can be orders of magnitude less than importance sampling [20]. This optimality result has been extended to a sequence of distributions $K > 1$ [29, eq. 16]. We further show that in the limit of a continuous path, the gap between annealed IS and annealed NCE is closed and we provide their estimation error:

**Theorem 1** (Estimation error and the Fisher-Rao path length) *For a finite value of $K$, the optimal loss is NCE:*

$$\mathrm{MSE}(p_0, p_1; q, K, N, \phi_{\mathrm{NCE}}) \leq \mathrm{MSE}(p_0, p_1; q, K, N, \phi), \qquad \forall q, K, N, \phi \ . \tag{7}$$

*In the limit of $K \to \infty$ (such that $K^2/N \to 0$), NCE, IS, and revIS converge to the same estimation error, given by the Fisher-Rao path length from the proposal to the target:*

$$\mathrm{MSE}(p_0, p_1; q, K, N, \phi) = \frac{1}{N} \int_0^1 I(t)dt + o\left(\frac{1}{N}\right) + o\left(\frac{K^2}{N}\right), \forall \phi \in \{\phi_{\mathrm{NCE}}, \phi_{\mathrm{IS}}, \phi_{\mathrm{RevIS}}\} \ , \tag{8}$$

*where the Fisher-Rao metric $I(t) := \mathbb{E}_{\boldsymbol{x} \sim p(\boldsymbol{x},t)}[(\frac{d}{dt} \log p_t(\boldsymbol{x}))^2]$ is defined as the Fisher information over the path, using time $t$ as the parameter.*

This is proven in Appendix B. Note that in this theorem, to take limits successively in $N \to \infty$ then in $K \to \infty$, we assume that $N$ grows at least as fast as $K^2$. While the NCE estimator requires solving a (potentially non-convex) scalar optimization problem in Eq. 2 and IS does not, this is the price to pay for statistical optimality. In the following, we will keep the optimal NCE loss and will indicate the dependency of the estimation error on $\phi_{\mathrm{NCE}}$ with a subscript, instead. We highlight that our theorems in this paper apply to the MSE in the limit of $K \to \infty$: their results hold the same for the IS and RevIS losses by virtue of theorem 1. Just as in the binary case, while the estimator is computed with the *unnormalized* path of densities (Eq. 2), the estimation error depends on the *normalized* path of densities (Eq. 8).

## 4.2 Number of distributions, $K + 1$

It is known that estimating the normalization constant using plain importance sampling ($K = 1$) can produce a statistical error that is exponential in the distance between the target and the proposal [30]. We show that in the binary case, NCE also suffers from an estimation error that scales exponentially with the parameter-distance between the target and proposal dimension.

In the following, we consider a proposal $p_0$ and target $p_1$ that are in an exponential family with sufficient statistics $\boldsymbol{t}(\boldsymbol{x})$. Note that exponential families have a rich history in statistical literature: they are classic parametric families of distributions [31] and certain of them have universal approximation capabilities [32, 33]. An exponential family is defined as

$$p(\boldsymbol{x}; \boldsymbol{\theta}) := \exp(\langle \boldsymbol{\theta}, \boldsymbol{t}(\boldsymbol{x}) \rangle - \log Z(\boldsymbol{\theta})) \tag{9}$$

where $Z(\boldsymbol{\theta}) = \int \exp(\langle \boldsymbol{\theta}_1, \boldsymbol{t}(\boldsymbol{x}) \rangle)$ is the partition function. We will consider that the (unnormalized) target density $f_1$ is what we call a *simply unnormalized model* defined as

$$f_1(\boldsymbol{x}) = \exp(\langle \boldsymbol{\theta}_1, \boldsymbol{t}(\boldsymbol{x}) \rangle) \tag{10}$$

Note that in general, a pdf can be unnormalized in many ways: one can multiply an unnormalized density by any positive function of $\boldsymbol{\theta}$ and it will still be unnormalized. However, the simple and intuitive case defined above is what we base the analysis below on.

For exponential families, the log-normalization $\log Z(\boldsymbol{\theta})$ is a convex function ("log-sum-exp") of the parameter $\boldsymbol{\theta}$ [34], which implies $0 \preccurlyeq \nabla_{\boldsymbol{\theta}}^2 \log Z(\boldsymbol{\theta})$. In our theorems, we use the further assumptions of strong convexity with constant $M$, and/or smoothness with constant $L$ (gradient is $L$-Liptschitz):

$$\nabla_{\boldsymbol{\theta}}^2 \log Z(\boldsymbol{\theta}) \succcurlyeq M \, \mathbf{Id} \tag{11}$$

$$\nabla_{\boldsymbol{\theta}}^2 \log Z(\boldsymbol{\theta}) \preccurlyeq L \, \mathbf{Id} \tag{12}$$

For exponential families, the derivatives of the log partition function yield moments of the sufficient statistics, and the hessian $\nabla_{\boldsymbol{\theta}}^2 \log Z(\boldsymbol{\theta}) = \mathrm{Cov}_{\boldsymbol{x} \sim p}[\boldsymbol{t}(\boldsymbol{x})]$ is in fact the Fisher matrix. We can interpret our two assumptions: Eq. 11 can be seen as a form of "strong identifiability". Namely, positive-definiteness is required of the Fisher matrix, for the Maximum-Likelihood loss to have a unique minimum: we further assume a lower-bound on the smallest eigenvalue, which can be viewed as a strong identifiability condition. Eq. 12 can be interpreted as a bound on the second-order moments of the distribution $p(\boldsymbol{x}; \boldsymbol{\theta})$, which is equivalent to the variance in every direction being bounded, which will be the case for parameters in a bounded domain $\boldsymbol{\theta} \in \boldsymbol{\Theta}$. An example along with the proofs of the following Theorems 2 and 3, are provided in Appendix B.

**Theorem 2** (Exponential error of binary NCE) *Assume the proposal $p_0$ is from the normalized exponential family, while the (unnormalized) target $f_1$ is from the simply unnormalized exponential family (Eq. 10). The log-partition function $\log Z(\boldsymbol{\theta})$ is assumed to be strongly convex (Eq. 11). Then in the binary case $K = 1$, the estimation error of NCE is (at least) exponential in the parameter-distance between the proposal and the target:*

$$\mathrm{MSE}_{\mathrm{NCE}}(p_0, p_1; q, K, N) \geq \frac{4}{N} \exp\left( \frac{1}{8} M \|\boldsymbol{\theta}_1 - \boldsymbol{\theta}_0\|^2 \right) - 1 + o\left( \frac{1}{N} \right), \qquad \text{when } K = 1 \tag{13}$$

*where $M$ is the strong convexity constant of $\log Z(\boldsymbol{\theta})$.*

Annealing the importance sampling estimator (increasing $K$) was proposed in the hope that we can trade the statistical cost in the dimension for a computational cost (number of classification tasks) which could be considered more acceptable. Yet, there is no definitive theory on the ability of annealing to reduce the statistical cost in a general setup [18, 21]. For both importance sampling and noise-contrastive estimation, we next prove that annealing with the near-universally used geometric path brings down the estimation error, from exponential to polynomial in the parameter-distance between the proposal and target. Given that we expect $\|\boldsymbol{\theta}_1 - \boldsymbol{\theta}_0\|_2$ to scale as $\sqrt{D}$ with the dimension, using these paths effectively makes annealed estimation amenable to high-dimensional problems. This corroborates empirical [35] and theoretical [17] results which suggested in simple cases that annealing with an appropriate path can reduce the estimator error up to several orders of magnitude.

**Theorem 3** (Polynomial error of annealed NCE with a geometric path) *Assume the proposal $p_0$ is from the normalized exponential family, while the (unnormalized) target $f_1$ is from the simply unnormalized exponential family (Eq. 10). The log-partition function $\log Z(\boldsymbol{\theta})$ is assumed to be strongly convex and smooth (Eq. 11, Eq. 12). Then in the annealing limit of a continuous path $K \to \infty$, the estimation error of annealed NCE with the geometric path is (at most) polynomial in the parameter-distance between the proposal and the target:*

$$\mathrm{MSE}_{\mathrm{NCE}}(p_0, p_1; q, K, N) \leq \frac{L^2}{MN} \|\boldsymbol{\theta}_1 - \boldsymbol{\theta}_0\|^2 + o\left( \frac{1}{N} \right) + o\left( \frac{K^2}{N} \right), \quad \text{when } q = 0 \tag{14}$$

*where $M$ and $L$ are respectively the strong convexity and smoothness constants of $\log Z(\boldsymbol{\theta})$.*

To our knowledge, this is the first result building on Gelman and Meng [17, Table 1] and Grosse et al. [18] which showcases the benefits of annealed estimation for a general target distribution.

We conclude that annealing with the near-universally used geometric path provably benefits noise-contrastive estimation, as well as importance sampling and reverse importance sampling, when the proposal and target distributions have little overlap.

## 4.3 Path parameter, $q$ — geometric vs. arithmetic

Despite the near-universal popularity of the geometric path ($q \to 0$), it is worth asking if there are other simple paths that are more optimal. Interpolating moments of exponential families was shown to outperform the geometric path by Grosse et al. [18], yet building such a path requires knowing the exponential family of the target. Other paths based on the arithmetic mean (and generalizations) of the target and proposal, were proposed in Masrani et al. [21], without a definitive theory of the estimation error.

Next, we analyze the error of the arithmetic path. We prove that the arithmetic path ($q = 1$) does *not* exhibit the same benefits as the geometric path: in general, its estimation error grows exponentially in the parameter-distance between the target and proposal distributions. However, in the case where an oracle gives us the normalization $Z_1$ to be used only in the construction of the path (we will discuss what this means in practice below), the arithmetic path can be reparameterized so as to bring down the estimation error to polynomial, even constant, in the parameter-distance. We start by the negative result.

**Theorem 4** (Exponential error of annealed NCE with an arithmetic path) *Assume the proposal $p_0$ is from the normalized exponential family, while the (unnormalized) target $f_1$ is from the simply unnormalized exponential family (Eq. 10). The log-partition function $\log Z(\boldsymbol{\theta})$ is assumed to be strongly convex (Eq. 11).*
*Consider the annealing limit of a continuous path $K \to \infty$ path and a far-away target with large enough $\|\boldsymbol{\theta}_1 - \boldsymbol{\theta}_0\| > 0$. For estimating the log normalization of the (unnormalized) target density $f_1$, the estimation error of annealed NCE with the arithmetic path is (at least) exponential in the parameter-distance between the proposal and the target:*

$$\text{MSE}_{\text{NCE}}(p_0, p_1; q, K, N) > \frac{C}{N} \exp\left(\frac{M}{2}\|\boldsymbol{\theta}_1 - \boldsymbol{\theta}_0\|^2\right) + o\left(\frac{1}{N}\right) + o\left(\frac{K^2}{N}\right), \quad \text{when } q = 1 \quad (15)$$

*where $C$ is constant defined in Appendix B.3.*

We suggest an intuitive explanation for this negative result. We begin with the observation that the estimation error (Eq. 8) depends on the *normalized* path of densities. Suppose the target model is rescaled by a constant 100, so that the new unnormalized target density is $f_1(\boldsymbol{x}) \times 100$ and its new normalization is $Z_1 \times 100$. Looking at table 2, this rescaling does not modify the geometric path of normalized densities, while it does the arithmetic path of normalized densities. Because the estimation error depends on path of normalized densities, this makes the arithmetic choice sensitive to target normalization, even more so as the parameter distance grows and the log-normalization with it, as a strongly convex function of it (Appendix, Eq. 94). This suggests making the arithmetic path of normalized distributions "robust" to the choice of $Z_1$. We will show this can be achieved by re-parameterizing the path in terms of $Z_1$.

We next prove that certain reparameterizations can bring down the error to a polynomial and even constant function of the parameter-distance between the target and proposal. The following theorems may seem purely theoretical, as if necessitating an oracle for $Z_1$, but they will actually lead to an efficient estimation algorithm later.

**Theorem 5** (Polynomial error of annealed NCE with an arithmetic path and oracle) *Assume the same as in Theorem 4, replacing the strong convexity of the log-partition by smoothness (Eq. 12). Additionally, suppose an oracle gives the normalization constant $Z_1$ to be used only in the reparameterization of the arithmetic path with $t \to \frac{t}{t + Z_1(1-t)}$ (see Table 2). This brings down the estimation error of annealed NCE to (at most) polynomial in the parameter-distance:*

$$\text{MSE}_{\text{NCE}}(p_0, p_1; q, K, N) \leq \frac{1}{N}(2 + L\|\boldsymbol{\theta}_1 - \boldsymbol{\theta}_0\|^2) + o\left(\frac{1}{N}\right) + o\left(\frac{K^2}{N}\right), \quad \text{when } q = 1 \quad (16)$$

*where $L$ is the smoothness constant of $\log Z(\boldsymbol{\theta})$.*

In fact, supposing we have (oracle) access to the normalizing constant $Z_1$, the arithmetic path can even be reparameterized such that it is the optimal path in a certain limit. We next prove such optimality in the limits of a continuous path $K \to \infty$ and "far-away" target and proposal [17]:

**Theorem 6** (Constant error of annealed NCE with an arithmetic path and oracle) *Consider the limits of a continuous annealing path $K \to \infty$, and of a target distribution whose overlap with the proposal*

| Path name | Unnormalized density | Normalized density | Error |
|---|---|---|---|
| Geometric | $f_t(\boldsymbol{x}) = p_0(\boldsymbol{x})^{1-t} f_1(\boldsymbol{x})^t$ | $p_t(\boldsymbol{x}) \propto p_0(\boldsymbol{x})^{1-t} p_1(\boldsymbol{x})^t$ | poly |
| Arithmetic | $f_t(\boldsymbol{x}) = (1 - w_t)p_0(\boldsymbol{x}) + w_t f_1(\boldsymbol{x})$ | $p_t(\boldsymbol{x}) = (1 - \tilde{w}_t)p_0(\boldsymbol{x}) + \tilde{w}_t p_1(\boldsymbol{x})$ | |
| vanilla | $w_t = t$ | $\tilde{w}_t = \frac{tZ_1}{(1-t)+tZ_1}$ | exp |
| oracle | $w_t = \frac{t}{t+Z_1(1-t)}$ | $\tilde{w}_t = t$ | poly |
| oracle-trig | $w_t = \frac{\sin^2\left(\frac{\pi t}{2}\right)}{\sin^2\left(\frac{\pi t}{2}\right)+Z_1 \cos^2\left(\frac{\pi t}{2}\right)}$ | $\tilde{w}_t = \sin^2\left(\frac{\pi t}{2}\right)$ | const |

Table 2: Geometric and arithmetic paths, defined in the space of unnormalized densities (second column); "oracle" and "oracle-trig" are reparameterizations of the arithmetic path which depend on the true normalization $Z_1$. The corresponding normalized densities (third column) produce an estimation error (fourth column) which we quantify.

*goes to zero. Namely, consider the quantities:*

$$\epsilon'(\boldsymbol{x}) := \sqrt{p_0(\boldsymbol{x})p_1(\boldsymbol{x})} \qquad\qquad \epsilon := \int_{\mathbb{R}^D} \epsilon'(\boldsymbol{x})d\boldsymbol{x} \qquad (17)$$

$$\epsilon''(\boldsymbol{x}) := \frac{\epsilon'(\boldsymbol{x}) - \epsilon \sin(\pi t)p_t^{\text{oracle}}(\boldsymbol{x})}{p_t^{\text{oracle-trig}}(\boldsymbol{x})} \quad \epsilon''' := \int_{\mathbb{R}^D} \int_0^1 \epsilon''(\boldsymbol{x})\left(1 + \frac{\left(\partial_t p_t^{\text{oracle-trig}}(\boldsymbol{x})\right)^2}{p_t^{\text{oracle-trig}}(\boldsymbol{x})}\right)dt d\boldsymbol{x} \ . \tag{18}$$

*Assume* $\sup_{\boldsymbol{x}\in\mathbb{R}^D} \epsilon'(x) \to 0, \sup_{\boldsymbol{x}\in\mathbb{R}^D} \epsilon''(x) \to 0, \epsilon \to 0, \epsilon''' \to 0$, *and consider the distributions* $p_t^{\text{arith-trig}}$ *and* $p_t^{\text{oracle}}$ *as defined in Table 2.*

*Then, the optimal annealing path convergences pointwise to an arithmetic path reparameterized trigonometrically with* $t \to \frac{t}{\sin^2(\frac{\pi t}{2})+Z_1(1-\sin^2(\frac{\pi t}{2}))}$ *and the estimation error tends to the optimal estimation error (which is constant with respect to the parameter-distance):*

$$\text{MSE}_{\text{NCE}}(p_0, p_1; q, K, N) = \frac{1}{N}\pi^2 + O\left(\frac{\epsilon + \epsilon'''}{N}\right) + o\left(\frac{1}{N}\right) + o\left(\frac{K^2}{N}\right) \quad \text{when } q = 1 \ . \tag{19}$$

By way of remarks, we note that the assumptions that the quantities $\sup \epsilon'(x), \sup \epsilon''(x), \epsilon, \epsilon'''$ go to 0 are mutually incomparable (i.e. none of them implies the others). For example, $\sup_{\boldsymbol{x}\in\mathbb{R}^D} \epsilon'(\boldsymbol{x})$ and $\epsilon$ going 0 require that the function $\sqrt{p_0(x)p_1(x)}$ goes to 0 in the $L_\infty$ and $L_1$ sense, respectively — and these two norms are not equivalent in the Lebesgue measure.

**Two-step estimation**   Thus, we see that, perhaps unsurprisingly, the "optimal" mixture weights in the space of unnormalized densities depends on the true $Z_1$: however, this dependency is simple. We propose a two-step estimation method: first, $Z_1$ is pre-estimated, for example using the geometric path; second, the estimate of $Z_1$ is plugged into the "oracle" or "oracle-trig" weight of the arithmetic path (table 2, column 2), and which is used to obtain a second estimation of $Z_1$. Note that pre-estimating a problematic (hyper)parameter, here $Z_1$, has proved beneficial to reduce the estimation error of NCE in a related context [36].

## 5   Numerical results

We now present numerical evidence for our theory and validate our two-step estimators. Importantly, we do *not* claim to achieve state of the art in terms of practical evaluation of the normalization constants; our goal is to support our theoretical analysis. We follow the evaluation methods of importance sampling literature [18] and evaluate our methods on synthetic Gaussians. This setup is specially convenient for validating our theory: the optimal estimation error can conveniently be computed in closed-form, so too can the geometric and arithmetic paths which avoids a sampling error from MCMC algorithms. These derivations are included in the Appendix B. We specifically consider the high-dimensional setting, where the computation of the determinant of a high-dimensional (covariance) matrix which appears in the normalization of a Gaussian, can in fact be challenging [37].

**Numerical Methods** The proposal distribution is always a standard Gaussian, while the target differs by the second moment: $p_1 = \mathcal{N}(\mathbf{0}, 2\,\mathbf{Id})$ in Figure 2, $p_1 = \mathcal{N}(\mathbf{0}, 0.25\,\mathbf{Id})$ in Figure 3b and $p_1 = \mathcal{N}(\mathbf{0}, \sigma^2\,\mathbf{Id})$ in Figure 3a, where the target variance decreases as $\sigma(i) = i^{-1}$ so that the (natural) parameter distance grows linearly [34, Part II-4]. We use a sample size of $N = 50000$ points, and, unless otherwise mentioned, $K + 1 = 10$ distributions from the annealing paths and the dimensionality is 50. To compute an estimator of the normalization constant using the non-convex NCE loss, we used a non-linear conjugate gradient scheme implemented in Scipy [38]. We chose the conjugate-gradient algorithm as it is deterministic (no residual variance like in SGD). The empirical Mean-Squared Error was computed over 100 random seeds, parallelized over 100 CPUs. The longest experiment was for Figure 3b and took 7 wall-clock hours to complete. For the two-step estimators ("two-step" and "two-step (trig)"), a pre-estimate of the normalization was first computed using the geometric path with 10 distributions. Then, this estimate was used to to re-parameterize an arithmetic path with 10 distributions which produced the second estimate.

**Results** Figure 2 numerically supports the optimality of the NCE loss for a finite $K$ (here, $K = 2$ so three distributions are used) proven in Theorem 1; note that the proposal distribution was normalized for this experiment, so that the log normalizing constant is zero. Figure 3 validates our main results for annealing paths. It shows how the estimation error scales with the proposal and target distributions growing apart, either with the parameter-distance in Figure 3a or with the dimensionality in Figure 3b. Using no annealing path ($K = 1$) produces an estimation error which grows linearly in log space; this numerically supports the exponential growth predicted by Theorem 2. Meanwhile, annealing ($K \to \infty$) sets the estimation error on different trends, depending on the choice of path. Choosing the geometric path brings the growth down to sub-exponential, as predicted by Theorem 3, while choosing the (basic) arithmetic path does not as in Theorem 4. To alleviate this, our two-step estimation methods consist in reparameterizing the arithmetic path so that it actually does bring down the estimation error. In fact, our two-step estimators in table 2 empirically approach the optimal estimation error in Figure 3. While this requires more computation, it has the appeal of making the estimation error *constant* with respect to the parameter-distance between the target and proposal distributions. Practically, this means that in Figure 3a, regular Noise-Contrastive Estimation (black, full line) fails when the parameter-distance between the target and proposal distributions is higher than 20, while our two-step estimators remain optimal.

We next explain interesting observations in Figure 3 which are actually coherent with our theory. First, in Figure 3a, the "two-step (trig)" estimator is only optimal when the parameter-distance between the target and proposal distributions is larger than 10. This is because the optimality of this two-step estimator was derived in Theorem 6 conditionally on non-overlapping distributions, here achieved by a large parameter-distance. Second, in both Figures 3a and 3b, the "two-step" estimator empirically achieves the optimal estimation error that was predicted for the "two-step (trig)" estimator. This suggests our polynomial upper bound from Theorem 5 may be loose in certain cases. This further explains why, in Figure 3a, the arithmetic path is near-optimal for a single value of the parameter-distance. At this value of 20, the partition function happens to be equal to one $Z(\theta_1) = 1$, so that the arithmetic path is effectively the same as the "two-step" estimator.

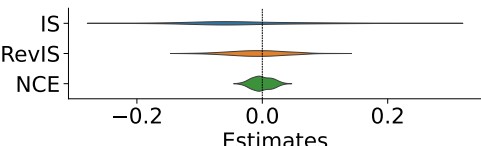

Figure 2: Optimality of the NCE loss, using the geometric path with $K = 2$. NCE has the smallest deviation from zero, the true value of the log normalizing constant.

# 6 Related work and Limitations

Previous work has mainly focused on annealed importance sampling [18, 39], which is a special case of our annealed Bregman estimator. They have evaluated the merits of different paths empirically, using an approximation of the estimation error called Effective Sample Size (ESS) and the consistency-

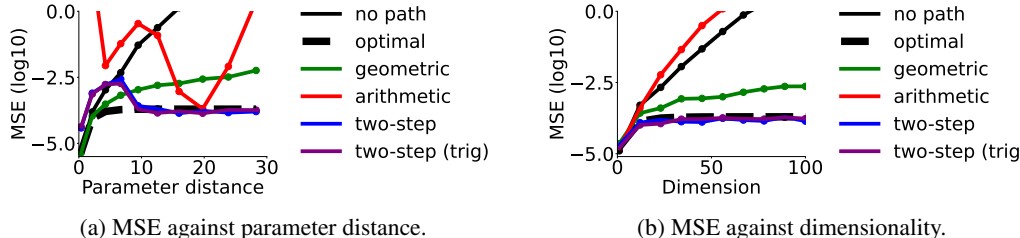

(a) MSE against parameter distance.  (b) MSE against dimensionality.

Figure 3: Estimation error as the target and proposal distributions grow apart. Without annealing, the error is exponential in the parameter distance (diagonal in log-scale). Annealing with the geometric path and our two-step methods brings down the error to slower growth, as predicted by our theorems.

gap. In our analysis, we consider consistent estimators and derive and optimize the exact estimation error of the optimal Noise-Contrastive Estimation. Liu et al. [27] considered the NCE estimate for $Z$ (not $\log Z$) with the name "discriminance sampling", and annealed the estimator using an extended state-space construction similar to Neal [10]. Their analysis of the estimation error is relevant but does not deal with hyperparameters other than the classification loss. Other annealed estimators also include annealed NCE as a special case but with little theory on its estimation error [17, 40, 29].

We made the common assumption of perfect sampling [17, 18, 41, 42] in order to make the estimation error tractable and obtain practical guidelines to reduce it. We note however, that this leaves a gap to bridge with a practical setup where the sampling error cannot be ignored; in fact, annealed importance sampling [10] was originally proposed such that the samples can be obtained from a Markov Chain that has not converged. In this original formulation, AIS is a special case of a larger framework called Sequential Monte Carlo (SMC) [43] in which the path of distributions is implicitly defined (by Markov transitions), sometimes even "on the go" [42]. Yet even within that theory, it seems that analyzing the estimation error for an inexplicit path of distributions is challenging [44, Eq. 38]. In particular, samples from MCMC will typically follow marginal distributions that are not analytically tractable, thus the stronger assumption of "perfect sampling" is often used to make estimation error depend explicitly on the path of distributions [45, Eq. 3.2]. Even then, results on how the estimation error scales with dimensionality are heuristic [10] or limited by assumptions such as an essentially log-concave path of distributions or a factorial target distribution [45].

It might also be argued that the limit of almost no overlap between proposal and target, which we use a lot, is unrealistic. To see why it can be realistic, consider the case of natural image data. A typical proposal is Gaussian, since nothing much more sophisticated is tractable in high dimensions. However, there is almost no overlap between Gaussian data and natural images, which is seen in the fact that a human observer can effortlessly discriminate between the two.

More generally, many methods based on "annealing" were developed to deal with sampling issues. In fact, the path costs for two such methods, parallel tempering [46, eq. 17] and tempered transitions [47, eq. 18], can be written with a sum of f-divergences which in the limit of a continuous path is the same cost function as in our Theorem 1. This suggests our results may apply to these methods as well.

## 7   Conclusion

We defined a class of estimators of the normalization constant, annealed Bregman estimation (ABE), which relies on a sampling phase from a path of distributions, and an estimation phase where these samples are used to estimate the log-normalization of the target distribution. Our results suggest a number of simple recommendations regarding hyperparameter choices in annealing. First, if the path has very few intermediate distributions, it is better to choose NCE due to its statistical optimality (Theorem 1). If however, the path has many intermediate distributions and approaches the annealing limit, then IS enjoys the same statistical optimality as NCE but has the advantage of its computational simplicity. Annealing can always provide substantial benefits (Theorem 2). Moreover, if we have a reasonable a priori estimate of $Z_1$, the arithmetic path achieves very low error (Theorem 5) — sometimes even approaching optimality (Theorem 6). On the other hand, even absent an initial estimate of $Z_1$, the geometric path can exponentially reduce the estimation error compared with no annealing (Theorems 2 and 3).

**Acknowledgements** This work was supported by the French ANR-20-CHIA-0016. Aapo Hyvärinen was supported by funding from the Academy of Finland and a Fellowship from CIFAR. Andrej Risteski was supported in part by NSF awards IIS-2211907, CCF-2238523, and an Amazon Research Award.

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

# Appendices

In the following, we will study the estimation error of annealed Bregman estimation (ABE) in two important setups: the log-normalization is computed using two distributions ($K = 1$), the proposal and the target, or else using a path of distributions ($K \to \infty$).

The code used for the experiments is available at `https://github.com/l-omar-chehab/annealing-normalizing-constants`.

## A No annealing, $K = 1$

We use [22, Eq.21] for the estimation error of any suitably parameterized [3] classifier $F(\boldsymbol{x}; \boldsymbol{\beta})$ between two distributions $p_1$ and $p_0$. The estimation error is measured by the asymptotic Mean-Squared Error (MSE)

$$\text{MSE}_{\hat{\boldsymbol{\beta}}}(p_n, \nu, \phi, N) = \frac{\nu + 1}{N}\text{tr}(\boldsymbol{\Sigma}) + o\left(\frac{1}{N}\right) \tag{20}$$

which depends on the sample sizes $N = N_1 + N_0$, their ratio $\nu = N_1/N_0$, the Bregman classification loss indexed by the convex function $\phi(x)$, and the asymptotic variance matrix

$$\boldsymbol{\Sigma} = \boldsymbol{I}_w^{-1}\left(\boldsymbol{I}_v - (1 + \frac{1}{\nu})\boldsymbol{m}_w \boldsymbol{m}_w^\top\right)\boldsymbol{I}_w^{-1} \ . \tag{21}$$

We suppose the standard technical conditions of van der Vaart [24, Th. 5.23] apply so that the remainder term $\|\hat{\boldsymbol{\beta}} - \boldsymbol{\beta}^*\|^2$ can indeed be written independently of the parameterization, as $o(N^{-1})$. Here, $\boldsymbol{m}_w(\boldsymbol{\beta}^*)$, $\boldsymbol{I}_w(\boldsymbol{\beta}^*)$ and $\boldsymbol{I}_v(\boldsymbol{\beta}^*)$ are the reweighted mean and covariances of the parameter-gradient of the classifier, also known as the "relative" Fisher score $\nabla_{\boldsymbol{\beta}} F(\boldsymbol{x}; \beta^*)$,

$$\boldsymbol{m}_w(\boldsymbol{\beta}^*) = \mathbb{E}_{\boldsymbol{x} \sim p_d}\left[w(\boldsymbol{x})\nabla_{\boldsymbol{\beta}} F(\boldsymbol{x}; \beta^*)\right] \tag{22}$$

$$\boldsymbol{I}_w(\boldsymbol{\beta}^*) = \mathbb{E}_{\boldsymbol{x} \sim p_d}\left[w(\boldsymbol{x})\nabla_{\boldsymbol{\beta}} F(\boldsymbol{x}; \beta^*)\nabla_{\boldsymbol{\beta}} F(\boldsymbol{x}; \beta^*)^\top\right] \tag{23}$$

$$\boldsymbol{I}_v(\boldsymbol{\beta}^*) = \mathbb{E}_{\boldsymbol{x} \sim p_d}\left[v(\boldsymbol{x})\nabla_{\boldsymbol{\beta}} F(\boldsymbol{x}; \beta^*)\nabla_{\boldsymbol{\beta}} F(\boldsymbol{x}; \beta^*)^\top\right] \tag{24}$$

where the reweighting of data points is by $w(\boldsymbol{x}) := \frac{p_1}{\nu p_0}(\boldsymbol{x})\phi''\left(\frac{p_1}{\nu p_0}(\boldsymbol{x})\right)$ and by $v(\boldsymbol{x}) = w(\boldsymbol{x})^2\frac{\nu p_0(\boldsymbol{x}) + p_1(\boldsymbol{x})}{\nu p_0(\boldsymbol{x})}$, which are all evaluated at the true parameter value $\boldsymbol{\beta}^*$.

**Scalar parameterization** We now consider a specific parameterization of the classifier:

$$F(\boldsymbol{x}; \beta) = \log\left(\frac{f_1(\boldsymbol{x})}{\nu f_0(\boldsymbol{x})}\right) - \beta \tag{25}$$

where the optimal parameter is the log-ratio of normalizations $\beta^* = \log(Z_1/Z_0)$. Consequently, we have $\nabla_{\boldsymbol{\beta}} F(\boldsymbol{x}; \beta^*) = -1$ and plugging this into the above quantities yields

$$\text{MSE} = \frac{1 + \nu}{T}\left(\frac{\mathbb{E}_{\boldsymbol{x} \sim p_1}\left[w^2(\boldsymbol{x})\frac{\nu p_0(\boldsymbol{x}) + p_1(\boldsymbol{x})}{\nu p_0(\boldsymbol{x})}\right]}{\mathbb{E}_{\boldsymbol{x} \sim p_1}[w(\boldsymbol{x})]^2} - (1 + \frac{1}{\nu})\right) + o\left(\frac{1}{N}\right)$$

which matches the formula found in [20, Eq 3.2]. For different choices of the Bregman classification loss, the estimation error is written using a divergence between the two distributions

| Name | Loss identified by $\phi(\boldsymbol{x})$ | Estimator | MSE, up to $o(N^{-1})$ |
|------|------|------|------|
| IS | $x \log x$ | $\log \mathbb{E}_{p_0} \frac{f_1}{f_0}$ | $\frac{1+\nu}{\nu N}\mathcal{D}_{\chi^2}(p_1, p_0)$ |
| RevIS | $-\log x$ | $-\log \mathbb{E}_{p_1} \frac{f_0}{f_1}$ | $\frac{1+\nu}{N}\mathcal{D}_{\chi^2}(p_0, p_1))$ |
| NCE | $x \log x - (1 + x)\log(\frac{1+x}{2})$ | implicit | $\frac{(1+\nu)^2}{\nu N}\frac{\mathcal{D}_{\text{HM}}(p_1, p_0)}{1 - \mathcal{D}_{\text{HM}}(p_1, p_0)}$ |
| IS-RevIS | $(1 - \sqrt{x})^2$ | $\log \mathbb{E}_{p_0} \frac{f_1}{f_0} - \log \mathbb{E}_{p_1} \frac{f_0}{f_1}$ | $\frac{(1+\nu)^2}{\nu N}\frac{1 - (1 - \mathcal{D}_{H^2}(p_d, p_n))^2}{(1 - \mathcal{D}_{H^2}(p_d, p_n))^2}$ |

---

[3] technically, the formula was derived in [48, 3] assuming the classifier was parameterized as $F(\boldsymbol{x}; \boldsymbol{\beta}) = \log p_1(\boldsymbol{x}; \boldsymbol{\beta})/\nu p_0(\boldsymbol{x})$ but the proof seems to generalize to any well-defined parameterization $F(\boldsymbol{x}; \boldsymbol{\beta})$.

where

$\mathcal{D}_{\chi^2}(p_1, p_0) := \left( \int \frac{p_1^2}{p_0} \right) - 1$ is the chi-squared divergence

$\mathcal{D}_{H^2}(p_1, p_0) := 1 - \left( \int \sqrt{p_1 p_0} \right) \in [0, 1]$ is the squared Hellinger distance

$\mathcal{D}_{\mathrm{HM}}(p_1, p_0) := 1 - \int \left( \pi p_1^{-1} + (1 - \pi) p_0^{-1} \right)^{-1} = 1 - \frac{1}{\pi} \mathbb{E}_{p_1} \frac{\pi p_0}{(1-\pi)p_1 + \pi p_0} \in [0, 1]$
is the harmonic divergence with weight $\pi \in [0, 1]$.
Here, the weight $\pi = P(Y = 0) = \frac{T_n}{T} = \frac{\nu}{1+\nu}$.

**Proof of Theorem 2** *Exponential error of binary NCE*

In the following, we will drop the remainder term in $o(N^{-1})$ given that no other limits are taken and that we will study the dominant term only. The estimation error of binary NCE is expressed in terms of the harmonic divergence

$$\mathrm{MSE} = \frac{4}{N} \frac{\mathcal{D}_{\mathrm{HM}}(p_1, p_0)}{1 - \mathcal{D}_{\mathrm{HM}}(p_1, p_0)} \tag{26}$$

which is intractable for general exponential families. Instead, we can lower-bound the estimation error. To do so, we lower-bound the harmonic divergence using the inequality of means (harmonic vs. geometric)

$$\mathcal{D}_{\mathrm{HM}}(p_1, p_0) = 1 - \int \frac{2 p_0 p_1}{p_0 + p_1} \geq 1 - \int \sqrt{p_0 p_1} = \mathcal{D}_{H^2}(p_0, p_1) \tag{27}$$

and therefore

$$\mathrm{MSE_{LB}} = \frac{4}{N} \frac{\mathcal{D}_{H^2}(p_1, p_0)}{1 - \mathcal{D}_{H^2}(p_1, p_0)} \ . \tag{28}$$

This lower bound is expressed in terms of the squared Hellinger distance, that is tractable for exponential families:

$$\mathcal{D}_{H^2}(p_1, p_0) := 1 - \int_{\boldsymbol{x} \in \mathbb{R}^D} \sqrt{p_1 p_0} d\boldsymbol{x} \tag{29}$$

$$= 1 - \int_{\boldsymbol{x} \in \mathbb{R}^D} \frac{1}{Z(\boldsymbol{\theta}_1)^{\frac{1}{2}} Z(\boldsymbol{\theta}_0)^{\frac{1}{2}}} \exp\left( \frac{1}{2} (\boldsymbol{\theta}_1 + \boldsymbol{\theta}_0)^\top \boldsymbol{t}(\boldsymbol{x}) \right) d\boldsymbol{x} \tag{30}$$

$$= 1 - \frac{Z(\frac{1}{2}\boldsymbol{\theta}_1 + \frac{1}{2}\boldsymbol{\theta}_0)}{Z(\boldsymbol{\theta}_1)^{\frac{1}{2}} Z(\boldsymbol{\theta}_0)^{\frac{1}{2}}} \tag{31}$$

$$= 1 - \exp\left( \log Z\left( \frac{1}{2}\boldsymbol{\theta}_1 + \frac{1}{2}\boldsymbol{\theta}_0 \right) - \frac{1}{2} \log Z(\boldsymbol{\theta}_1) - \frac{1}{2} \log Z(\boldsymbol{\theta}_0) \right) \ . \tag{32}$$

We now wish to lower bound $\mathrm{MSE_{LB}}$, and therefore $\mathcal{D}_{H^2}(p_1, p_0)$, by an expression which is exponential in the parameter distance $\|\boldsymbol{\theta}_1 - \boldsymbol{\theta}_0\|$. To do so, we note that for exponential families, the log-normalization is convex in the parameters. Here, we further assume strong convexity, so that

$$\log Z\left( \frac{1}{2}\boldsymbol{\theta}_1 + \frac{1}{2}\boldsymbol{\theta}_0 \right) \leq \frac{1}{2} \log Z(\boldsymbol{\theta}_1) + \frac{1}{2} \log Z(\boldsymbol{\theta}_0) - \frac{1}{8} M \|\boldsymbol{\theta}_1 - \boldsymbol{\theta}_0\|^2 \tag{33}$$

where $M$ is the strong convexity constant. Plugging this back into the squared Hellinger distance, we obtain

$$\mathcal{D}_{H^2}(p_1, p_0) \geq 1 - \exp\left( -\frac{1}{8} M \|\boldsymbol{\theta}_1 - \boldsymbol{\theta}_0\|^2 \right) \tag{34}$$

so that the MSE

$$\mathrm{MSE} \geq \frac{4}{N} \frac{\mathcal{D}_{H^2}(p_1, p_0)}{1 - \mathcal{D}_{H^2}(p_1, p_0)} \geq \frac{4}{N} \exp\left( \frac{1}{8} M \|\boldsymbol{\theta}_1 - \boldsymbol{\theta}_0\|^2 \right) - 1 \tag{35}$$

grows exponentially with the euclidean distance between the parameters.

# B   Annealing limit, $K \to \infty$

We now consider annealing paths $(p_t)_{t \in [0,1]}$ that interpolate between between the proposal $p_0$ and the target $p_1$.

## B.1   Estimation error

We first show the optimality of the NCE loss within the family of Annealed Bregman Estimators, in the sense that it produces the smallest estimation error. We then study the estimation error of different annealed Bregman estimators in the annealing limit of a continuous path ($K \to \infty$).

**Proof of Theorem 1**   *Optimality of the NCE loss and the estimation error in the annealing limit* $K \to \infty$

- Optimality of the NCE loss

  Because the annealed Bregman estimator is built by adding independent estimators

$$\widehat{\log Z_1} = \sum_{k=0}^{K-1} \log \left( \widehat{\frac{Z_{(k+1)/K}}{Z_{k/K}}} \right) + \log Z_0 \quad . \tag{36}$$

  the total Mean Squared Error (MSE) is the sum of each MSEs for each estimator (indexed by $k \in [\![0, K-1]\!]$)

$$\mathrm{MSE}((\phi_k)_{k \in [\![0,K]\!]}) = \sum_{k=0}^{K-1} \mathrm{MSE}_k(\phi_k) \tag{37}$$

  where we highlighted the dependency on the classification losses identified by $(\phi_k)_{k \in [\![0,K]\!]}$. The MSEs follow Eq. 26. It was shown by Meng and Wong [20] that for any of these MSEs, the optimal loss is identified by $\phi_k(x) = x \log x - (1+x) \log(\frac{1+x}{2})$ and is in fact the NCE loss [22]. Thus the sum of MSEs is minimized for the same loss.

- Annealed Noise-Contrastive Estimation (NCE)

  We are interested in the estimation error (asymptotic MSE) obtained for the NCE loss. Based off table 1, it is written as

$$\mathrm{MSE} = \sum_{k=0}^{K-1} \left( \frac{4K}{N} \frac{\mathcal{D}_{\mathrm{HM}}(p_{k/K}, p_{(k+1)/K})}{1 - \mathcal{D}_{\mathrm{HM}}(p_{k/K}, p_{(k+1)/K})} + o\left(\frac{K}{N}\right) \right) \tag{38}$$

$$= \frac{4K}{N} \sum_{k=0}^{K-1} \frac{\mathcal{D}_{\mathrm{HM}}(p_{k/K}, p_{(k+1)/K})}{1 - \mathcal{D}_{\mathrm{HM}}(p_{k/K}, p_{(k+1)/K})} + o\left(\frac{K^2}{N}\right) \quad . \tag{39}$$

  where a sample budget of $N/K$ is used for each estimator that is summed. The estimation error of balanced ($\nu = 1$) NCE-JS between two "close" distributions $p_t$ and $p_{t+h}$, is

$$\mathrm{MSE}(p_t, p_{t+h}) \propto \frac{\mathcal{D}_{\mathrm{HM}}(p_t, p_{t+h})}{1 - \mathcal{D}_{\mathrm{HM}}(p_t, p_{t+h})} \tag{40}$$

  up to the remainder term. The estimation error can be simplified using a Taylor expansion. To do so, we recall that $D_{\mathrm{HM}}$ is an f-divergence generated by $\phi(x) = 1 - \frac{x}{\pi + (1-\pi)x}$ [49, 50] ($\pi = \frac{1}{2}$ here) and its expansion is therefore [51, Eq.7.64]

$$\mathcal{D}_{\mathrm{HM}}(p_t, p_{t+h}) = \frac{1}{2} h^2 \nabla_t^2 \mathcal{D}_{\mathrm{HM}}(p_t, p_{t+h}) + o(h^2) \tag{41}$$

$$= \frac{1}{2} \phi''(1) h^2 I(t) + o(h^2) = \frac{1}{4} h^2 I(t) + o(h^2) \quad . \tag{42}$$

  It follows that

$$\frac{\mathcal{D}_{\mathrm{HM}}(p_t, p_{t+h})}{1 - \mathcal{D}_{\mathrm{HM}}(p_t, p_{t+h})} = \frac{1}{4} I(t) h^2 + o(h^2) \quad . \tag{43}$$

Summing these estimation errors along the path of distributions with $h = 1/K$,

$$\text{MSE} = \frac{4K}{N} \sum_{k=0}^{K-1} \left( \frac{1}{4} I(t) \frac{1}{K^2} + o\left(\frac{1}{K^2}\right) \right) + o\left(\frac{K^2}{N}\right) \tag{44}$$

$$= \left( \frac{1}{NK} \sum_{k=0}^{K-1} I(t) \right) + o\left(\frac{1}{N}\right) + o\left(\frac{K^2}{N}\right) \tag{45}$$

$$= \frac{1}{N} \int_0^1 I(t) dt + o\left(\frac{1}{N}\right) + o\left(\frac{K^2}{N}\right) \quad . \tag{46}$$

In the case of a parametric path $p(\boldsymbol{x}|\boldsymbol{\theta}(t))_{t \in [0,1]}$, the proof is the same. Simply, the second-order term in the Taylor expansion of Eq. 42 is computed using the chain rule

$$\nabla_t^2 \text{MSE}(p_{\boldsymbol{\theta}(t)}, p_{\boldsymbol{\theta}(t+h)}) \tag{47}$$

$$= \dot{\boldsymbol{\theta}}(t)^\top \nabla_{\boldsymbol{\theta}}^2 \text{MSE}(p_{\boldsymbol{\theta}(t)}, p_{\boldsymbol{\theta}(t+h)}) \dot{\boldsymbol{\theta}}(t) + \ddot{\boldsymbol{\theta}}(t)^\top \nabla_{\boldsymbol{\theta}} \text{MSE}(p_{\boldsymbol{\theta}(t)}, p_{\boldsymbol{\theta}(t+h)}) \tag{48}$$

$$= \dot{\boldsymbol{\theta}}(t)^\top \nabla_{\boldsymbol{\theta}}^2 \text{MSE}(p_{\boldsymbol{\theta}(t)}, p_{\boldsymbol{\theta}(t+h)}) \dot{\boldsymbol{\theta}}(t)^\top + 0 \tag{49}$$

$$= \dot{\boldsymbol{\theta}}(t)^\top \boldsymbol{I}(\theta(t)) \dot{\boldsymbol{\theta}}(t) \tag{50}$$

- Annealed importance sampling (IS)

  Similarly, for the choice of the importance sampling base estimator,

$$\text{MSE} = \sum_{k=0}^{K-1} \left( \frac{2K}{N} \mathcal{D}_{\chi^2}(p_{(k+1)/K}, p_{k/K}) + o\left(\frac{K}{N}\right) \right) \tag{51}$$

$$= \frac{2K}{N} \sum_{k=0}^{K-1} \mathcal{D}_{\chi^2}(p_{(k+1)/K}, p_{k/K}) + o\left(\frac{K^2}{N}\right) \tag{52}$$

$$= \frac{2K}{N} \sum_{k=0}^{K-1} \mathcal{D}_{\text{rev}\chi^2}(p_{k/K}, p_{(k+1)/K}) + o\left(\frac{K^2}{N}\right) \tag{53}$$

$$= \frac{2K}{N} \sum_{k=0}^{K-1} \left( \frac{1}{2} \phi''(1) I(t) \frac{1}{K^2} + o\left(\frac{1}{K^2}\right) \right) + o\left(\frac{K^2}{N}\right) \tag{54}$$

$$= \frac{1}{NK} \sum_{k=0}^{K-1} \phi''(1) I(t) + o\left(\frac{1}{N}\right) + o\left(\frac{K^2}{N}\right) \tag{55}$$

$$= \frac{1}{N} \int_0^1 I(t) dt + o\left(\frac{1}{N}\right) + o\left(\frac{K^2}{N}\right) . \tag{56}$$

given that $\phi(x) = -\log(x)$ and therefore $\phi''(1) = 1$ for the reverse $\chi^2$ divergence.

- Annealed reverse importance sampling (RevIS)

  Similarly, for the choice of the reverse importance sampling base estimator,

$$\text{MSE} = \sum_{k=0}^{K-1} \left( \frac{2K}{N} \mathcal{D}_{\chi^2}(p_{k/K}, p_{(k+1)/K}) + o\left(\frac{K}{N}\right) \right) \tag{57}$$

$$= \frac{2K}{N} \sum_{k=0}^{K-1} \left( \frac{1}{2} \phi''(1) I(t) \frac{1}{K^2} + o\left(\frac{1}{K^2}\right) \right) + o\left(\frac{K^2}{N}\right) \tag{58}$$

$$= \frac{1}{NK} \sum_{k=0}^{K-1} I(t) + o\left(\frac{1}{N}\right) + o\left(\frac{K^2}{N}\right) = \frac{1}{N} \int_0^1 I(t) dt + o\left(\frac{1}{N}\right) + o\left(\frac{K^2}{N}\right) . \tag{59}$$

given that $\phi(x) = x \log(x)$ and therefore $\phi''(1) = 1$ for the $\chi^2$ divergence.

## B.2 Examples of paths

**Geometric path**  The geometric path is defined in the space of unnormalized densities by

$$f_t(\boldsymbol{x}) := p_0(\boldsymbol{x})^{1-t} f_1(\boldsymbol{x})^t = p_0(\boldsymbol{x})^{1-t} p_1(\boldsymbol{x})^t Z_1^t \propto p_0(\boldsymbol{x})^{1-t} p_1(\boldsymbol{x})^t \tag{60}$$

so in the space of normalized densities, the path is

$$p_t := \frac{p_0(\boldsymbol{x})^{1-t} p_1(\boldsymbol{x})^t}{Z_t} \tag{61}$$

where the normalization is

$$Z_t := \int_{\boldsymbol{x} \in \mathbb{R}^d} p_0(\boldsymbol{x})^{1-t} p_1(\boldsymbol{x})^t d\boldsymbol{x} = \mathbb{E}_{\boldsymbol{x} \sim p_1}\left[\left(\frac{p_0(\boldsymbol{x})}{p_1(\boldsymbol{x})}\right)^t\right] = \mathbb{E}_{\boldsymbol{x} \sim p_0}\left[\left(\frac{p_1(\boldsymbol{x})}{p_0(\boldsymbol{x})}\right)^{1-t}\right] . \tag{62}$$

**Arithmetic path**  The arithmetic path is defined in the space of unnormalized densities by

$$f_t(\boldsymbol{x}) := (1-t)p_0(\boldsymbol{x}) + tf_1(\boldsymbol{x}) = (1-t)p_0 + tZ_1 p_1 \tag{63}$$

$$\propto \frac{(1-t)}{(1-t) + tZ_1} p_0 + \frac{tZ_1}{(1-t) + tZ_1} p_1 \tag{64}$$

so in the space of normalized densities, the path is actually a mixture between the target and the proposal, where the weight of the mixture is a nonlinear function of the target normalization

$$p_t := (1 - \tilde{w}_t)p_0 + \tilde{w}_t p_1, \quad \tilde{w}_t = \frac{tZ_1}{(1-t) + tZ_1} . \tag{65}$$

**Optimal path**  We know (*e.g.* from Gelman and Meng [17, Eq. 49]) that the optimal path is

$$p_t(\boldsymbol{x}) = \left(a(t)\sqrt{p_0(\boldsymbol{x})} + b(t)\sqrt{p_1(\boldsymbol{x})}\right)^2 \tag{66}$$

where the coefficients $a(t)$ and $b(t)$

$$a(t) = \frac{\cos((2t-1)\alpha_H)}{2\cos(\alpha_H)} - \frac{\sin((2t-1)\alpha_H)}{2\sin(\alpha_H)} \tag{67}$$

$$b(t) = \frac{\cos((2t-1)\alpha_H)}{2\cos(\alpha_H)} + \frac{\sin((2t-1)\alpha_H)}{2\sin(\alpha_H)} \tag{68}$$

are simple functions of the squared Hellinger distance $\mathcal{D}_{H^2}(p_0, p_1)$ between the proposal and the target [4]

$$\alpha_H = \arctan\left(\sqrt{\frac{\mathcal{D}_{H^2}(p_0, p_1)}{2 - \mathcal{D}_{H^2}(p_0, p_1)}}\right) \in [0, \frac{\pi}{4}] . \tag{69}$$

The estimation error produced by that optimal path is [17, Eq. 48]

$$\mathrm{MSE} = \frac{1}{N}\int_0^1 I(t)dt = \frac{1}{N}16\alpha_H^2 . \tag{70}$$

For two Gaussians

$$p_0 := \mathcal{N}(\boldsymbol{\mu}_0, \boldsymbol{\Sigma}_0) \tag{71}$$

$$p_1 := \mathcal{N}(\boldsymbol{\mu}_1, \boldsymbol{\Sigma}_1) \tag{72}$$

the squared Hellinger distance can be written in closed-form

$$\mathcal{D}_{H^2}(p_0, p_1) = 1 - \frac{|\boldsymbol{\Sigma}_0|^{\frac{1}{4}}|\boldsymbol{\Sigma}_1|^{\frac{1}{4}}}{|\frac{1}{2}\boldsymbol{\Sigma}_0 + \frac{1}{2}\boldsymbol{\Sigma}_1|^{\frac{1}{2}}} \exp\left(-\frac{1}{8}(\boldsymbol{\mu}_1 - \boldsymbol{\mu}_0)^\top \left(\frac{1}{2}\boldsymbol{\Sigma}_0 + \frac{1}{2}\boldsymbol{\Sigma}_1\right)^{-1}(\boldsymbol{\mu}_1 - \boldsymbol{\mu}_0)\right) \tag{73}$$

and plugs into the optimal path formula, which is also obtained in closed-form.

---

[4]In Gelman and Meng [17, Eq. 49], the Hellinger distance is defined such that it is in $[0, \sqrt{2}]$. We here instead use the conventional definition of the squared Hellinger distance which is normalized so that it is in $[0, 1]$.

## B.3 Estimation error from taking different paths

**Proof of Theorem 3** *Polynomial error of annealed NCE with the geometric path*

We next study the estimation error produced by the geometric path (Figure 1). In the annealing limit $K \to \infty$, the MSE is written as

$$\text{MSE} = \frac{1}{N} \int_0^1 I(t)dt + o\left(\frac{1}{N}\right) + o\left(\frac{K^2}{N}\right) . \tag{74}$$

We recall from Grosse et al. [18] that the geometric path is closed for distributions in the exponential family: all distributions along the path remain in the exponential family. Furthermore, their Fisher information can be written in terms of the terms parameters [18, Eq. 17]; this is based off a a result of exponential families from [52, Section 3.3]

$$I(t) = \dot{\boldsymbol{\theta}}(t)^\top \boldsymbol{I}(\boldsymbol{\theta}(t))\dot{\boldsymbol{\theta}}(t) = \dot{\boldsymbol{\theta}}(t)^\top \dot{\boldsymbol{\mu}}(t) \tag{75}$$

where $\boldsymbol{\mu}(t)$ are the generalized moments, defined as $\boldsymbol{\mu}(t) = \mathbb{E}_{\boldsymbol{x} \sim p_t(\boldsymbol{x})}[\boldsymbol{t}(\boldsymbol{x})] = \nabla_{\boldsymbol{\theta}} \log Z_t(\boldsymbol{\theta})$. It follows,

$$\frac{1}{N} \int_0^1 I(t)dt = \frac{1}{N} \int_0^1 \dot{\boldsymbol{\theta}}(t)^\top \dot{\boldsymbol{\mu}}(t)dt . \tag{76}$$

The geometric path is defined in parameter space by $\boldsymbol{\theta}_t = t\boldsymbol{\theta}_1 + (1-t)\boldsymbol{\theta}_0$, therefore

$$\frac{1}{N} \int_0^1 I(t)dt = \frac{1}{N}(\boldsymbol{\theta}_1 - \boldsymbol{\theta}_0)^\top \int_0^1 \dot{\boldsymbol{\mu}}(t)dt = \frac{1}{N}(\boldsymbol{\theta}_1 - \boldsymbol{\theta}_0)^\top (\boldsymbol{\mu}_1 - \boldsymbol{\mu}_0) \tag{77}$$

as in [18, Eq. 17]. For exponential families, $\log Z(\boldsymbol{\theta})$ is convex in $\boldsymbol{\theta}$. Here, we further assume strong convexity (with constant $M$) and smoothness (with constant $L$) so that

$$(\boldsymbol{\theta}_1 - \boldsymbol{\theta}_0)^\top (\boldsymbol{\mu}_1 - \boldsymbol{\mu}_0) = (\boldsymbol{\theta}_1 - \boldsymbol{\theta}_0)^\top (\nabla_{\boldsymbol{\theta}} \log Z_t(\boldsymbol{\theta}_1) - \nabla_{\boldsymbol{\theta}} \log Z_t(\boldsymbol{\theta}_0)) \tag{78}$$

$$\leq \frac{1}{M}\|\nabla_{\boldsymbol{\theta}} \log Z_t(\boldsymbol{\theta}_1) - \nabla_{\boldsymbol{\theta}} \log Z_t(\boldsymbol{\theta}_0)\|^2 \leq \frac{L^2}{M}\|\boldsymbol{\theta}_1 - \boldsymbol{\theta}_0\|^2 \tag{79}$$

so that the MSE

$$\text{MSE} \leq \frac{L^2}{MN}\|\boldsymbol{\theta}_1 - \boldsymbol{\theta}_0\|^2 + o\left(\frac{1}{N}\right) + o\left(\frac{K^2}{N}\right) \tag{80}$$

is polynomial in the euclidean distance between the parameters.

**Proof of Theorem 4** *Exponential error of annealed NCE with the arithmetic path and "vanilla" schedule*

We now study the estimation error produced by the arithmetic path with "vanilla" schedule (table 2, line 3). Similarly, we start with the formula of the estimation error of NCE in the limit of a continuous path

$$\text{MSE} = \frac{1}{N} \int_0^1 I(t)dt + o\left(\frac{1}{N}\right) + o\left(\frac{K^2}{N}\right) \tag{81}$$

where $I(t) = \mathbb{E}_{\boldsymbol{x} \sim p(\boldsymbol{x},t)}[(\frac{d}{dt} \log p(\boldsymbol{x},t))^2]$ is the Fisher information over the path, using time $t$ as the parameter. The arithmetic path is a Gaussian mixture (see table 2) so we will conveniently use the parametric form of the path to compute the Fisher information

$$I(t) = \dot{\tilde{w}}_t^\top I(\tilde{w}_t)\dot{\tilde{w}}_t \tag{82}$$

where the parameter here is the weight of the Gaussian mixture $\tilde{w}_t = tZ_1/(tZ_1 + 1 - t)$. We will need to compute two quantities: the Fisher information to that mixture parameter (not the time), and the parameter speed $\dot{\tilde{w}}_t$.

$$I(\tilde{w}_t) := \mathbb{E}_{\boldsymbol{x} \sim p_{\tilde{w}_t}}\left[\left(\frac{\partial \log p_{\tilde{w}_t}}{\partial \tilde{w}_t}(\boldsymbol{x})\right)^2\right] = \mathbb{E}_{\boldsymbol{x} \sim p_{\tilde{w}_t}}\left[\left(\frac{1}{p_{\tilde{w}_t}(\boldsymbol{x})}\frac{\partial p_{\tilde{w}_t}}{\partial \tilde{w}_t}(\boldsymbol{x})\right)^2\right] \tag{83}$$

$$= \int_{\boldsymbol{x} \in \mathbb{R}^D} \frac{(p_1(\boldsymbol{x}) - p_0(\boldsymbol{x}))^2}{p_{\tilde{w}_t}(\boldsymbol{x})}d\boldsymbol{x} = \int_{\boldsymbol{x} \in \mathbb{R}^D} \frac{(p_1(\boldsymbol{x}) - p_0(\boldsymbol{x}))^2}{(1 - \tilde{w}_t)p_0(\boldsymbol{x}) + \tilde{w}_t p_1(\boldsymbol{x})}d\boldsymbol{x} \tag{84}$$

$$\geq \int_{\boldsymbol{x} \in \mathbb{R}^D} \frac{(p_1(\boldsymbol{x}) - p_0(\boldsymbol{x}))^2}{p_0(\boldsymbol{x}) + p_1(\boldsymbol{x})}d\boldsymbol{x} = \int p_0(\boldsymbol{x})\frac{\left(1 - \frac{p_1(\boldsymbol{x})}{p_0(\boldsymbol{x})}\right)^2}{1 + \frac{p_1(\boldsymbol{x})}{p_0(\boldsymbol{x})}} = \mathcal{D}_\phi(p_1, p_0) \tag{85}$$

which is an f-divergence with generator $\phi(\boldsymbol{x}) = (1-x)^2/(1+x)$ that provides a $t$-independent lower bound. This will allow us to factor this quantity out of the integral defining the MSE, and simplify computations. We also have

$$\dot{\tilde{u}}_t := \frac{\partial}{\partial t}\tilde{w}_t = \frac{1}{t(1-t)} \times \sigma\left(\log \frac{tZ_1}{1-t}\right) \times \left(1 - \sigma\left(\log \frac{tZ_1}{1-t}\right)\right) \tag{86}$$

$$= \frac{1}{t(1-t)} \times \frac{tZ_1}{(1-t)+tZ_1} \times \frac{(1-t)}{(1-t)+tZ_1} = \frac{Z_1}{((1-t)+tZ_1)^2} \ . \tag{87}$$

where we choose to keep the dependency on $t$. The intuition is that integrating this quantity will yield a function of $Z_1$, which will drive the MSE toward high values. We next show this rigorously and finally compute the estimation error.

$$\frac{1}{N}\int_0^1 I(t)dt = \frac{1}{N}\int_0^1 \dot{\tilde{w}}(t)I(\tilde{w}(t))\dot{\tilde{w}}(t)dt \tag{88}$$

$$\geq \frac{1}{N} \times \mathcal{D}_\phi(p_1, p_0) \times \int_0^1 \dot{\tilde{w}}(t)^2 dt \tag{89}$$

$$= \frac{1}{N} \times \mathcal{D}_\phi(p_1, p_0) \times Z_1^2 \times \int_0^1 \frac{1}{(t(Z_1-1)+1)^4}dt \tag{90}$$

$$= \frac{1}{N} \times \mathcal{D}_\phi(p_1, p_0) \times Z_1^2 \times \frac{Z_1^2 + Z_1 + 1}{3Z_1^3} \tag{91}$$

$$= \frac{1}{3N} \times \mathcal{D}_\phi(p_1, p_0) \times (Z_1^{-1} + 1 + Z_1) \ . \tag{92}$$

We would like to write $Z_1$ in terms of the parameters. To do so, we now suppose the unnormalized target is in a simply unnormalized exponential family. Consequently,

$$Z_1 := \exp(\log Z(\boldsymbol{\theta}_1) - \log Z(\boldsymbol{\theta}_0) + \log Z(\boldsymbol{\theta}_0)) \tag{93}$$

$$\geq \exp\left(\nabla \log Z(\boldsymbol{\theta}_0)(\boldsymbol{\theta}_1 - \boldsymbol{\theta}_0) + \frac{M}{2}\|\boldsymbol{\theta}_1 - \boldsymbol{\theta}_0\|^2 + \log Z(\boldsymbol{\theta}_0)\right) \ . \tag{94}$$

using the strong convexity of the log-partition function. For large enough $\|\boldsymbol{\theta}_1 - \boldsymbol{\theta}_0\| > 0$, the quadratic term in the exponential is larger than the linear term, and the divergence $\mathcal{D}_\phi(p_1, p_0)$ is larger than a constant. It follows that for large enough $\|\boldsymbol{\theta}_1 - \boldsymbol{\theta}_0\| > 0$, there exists a constant $C > 0$ such that the MSE grows (at least) exponentially with the parameter-distance

$$\mathrm{MSE} > \frac{C}{3N} \times \exp\left(\frac{M}{2}\|\boldsymbol{\theta}_1 - \boldsymbol{\theta}_0\|^2\right) + o\left(\frac{1}{N}\right) + o\left(\frac{K^2}{N}\right) \ . \tag{95}$$

**Proof of Theorem 5** *Polynomial error of annealed NCE with the arithmetic path and "oracle" schedule*

We now study the estimation error produced by the arithmetic path with "oracle" schedule (table 2, line 4). Similarly, we start with the formula of the estimation error of NCE annealed over a continuous path

$$\mathrm{MSE} = \frac{1}{N}\int_0^1 I(t)dt + o\left(\frac{1}{N}\right) + o\left(\frac{K^2}{N}\right) \tag{96}$$

where $I(t) = \mathbb{E}_{\boldsymbol{x}\sim p(\boldsymbol{x},t)}[(\frac{d}{dt}\log p(\boldsymbol{x},t))^2]$ is the Fisher information over the path, using time $t$ as the parameter. The arithmetic path is the Gaussian mixture $p_t(\boldsymbol{x}) = tp_1(\boldsymbol{x}) + (1-t)p_0(\boldsymbol{x})$ (see table 2). The Fisher information is therefore

$$I(t) := \mathbb{E}_{\boldsymbol{x}\sim p_t}\left[\left(\frac{\partial \log p_t}{\partial t}(\boldsymbol{x})\right)^2\right] = \mathbb{E}_{\boldsymbol{x}\sim p_t}\left[\left(\frac{1}{p_t(\boldsymbol{x})}\frac{\partial p_t}{\partial t}(\boldsymbol{x})\right)^2\right] \tag{97}$$

$$= \int_{\boldsymbol{x}\in\mathbb{R}^D} \frac{(p_1(\boldsymbol{x}) - p_0(\boldsymbol{x}))^2}{p_t(\boldsymbol{x})}d\boldsymbol{x} = \int_{\boldsymbol{x}\in\mathbb{R}^D} \frac{(p_1(\boldsymbol{x}) - p_0(\boldsymbol{x}))^2}{(1-t)p_0(\boldsymbol{x}) + tp_1(\boldsymbol{x})}d\boldsymbol{x} \tag{98}$$

$$\leq \int_{\boldsymbol{x}\in\mathbb{R}^D} \frac{p_1(\boldsymbol{x})^2 + p_0(\boldsymbol{x})^2}{(1-t)p_0(\boldsymbol{x}) + tp_1(\boldsymbol{x})}d\boldsymbol{x} \tag{99}$$

where we choose to keep the dependency on $t$ in the bound.

We briefly justify this choice. We had first tried a $t$-independent bound, which led to an upper bound of the MSE that was too loose. We share insight as to why: first, recognize that the fraction can be broken in two terms, each of them a chi-square divergence between an endpoint of the path ($p_0$ or $p_1$) and the mixture $p_t$. Each of them admits a $t$-independent upper bound given by the chi-square divergence between the endpoints $p_0$ and $p_1$, using lemma 1. However, the chi-square divergence between two Gaussians, for example, is exponential (not polynomial) in the natural parameters [51, eq 7.41]. In fact, plotting $I(t)$ for a univariate Gaussian model revealed that it took high values at the endpoints $t = 0$ and $t = 1$, and was near zero almost everywhere else in the interval $t \in [0, 1]$, which again suggested that dropping the dependency on $t$ was unreasonable.

Now we can compute the estimation error, as

$$\frac{1}{N} \int_0^1 I(t)dt \le \frac{1}{N} \int_{\mathbb{R}^d} \int_0^1 \frac{p_1(\boldsymbol{x})^2 + p_0(\boldsymbol{x})^2}{(1-t)p_0(\boldsymbol{x}) + tp_1(\boldsymbol{x})} dtd\boldsymbol{x} = \frac{1}{N}(J_1 + J_2) \ . \tag{100}$$

Let us try to solve one of these integrals, say $J_1$.

$$J_1 = \int_{\mathbb{R}^d} \int_0^1 \frac{p_1(\boldsymbol{x})^2}{(1-t)p_0(\boldsymbol{x}) + tp_1(\boldsymbol{x})} dtd\boldsymbol{x} = \int_{\mathbb{R}^d} \frac{p_1(\boldsymbol{x})^2}{p_0(\boldsymbol{x})} \left( \int_0^1 \frac{1}{1 + t(\frac{p_1(\boldsymbol{x})}{p_0(\boldsymbol{x})} - 1)} dt \right) d\boldsymbol{x} \tag{101}$$

$$= \int_{\mathbb{R}^d} \frac{p_1(\boldsymbol{x})^2}{p_0(\boldsymbol{x})} \left( \frac{1}{\frac{p_1}{p_0} - 1} \log \frac{p_1}{p_0} \right) d\boldsymbol{x} = 1 + \mathbb{E}_{p_1} \left[ \frac{1}{1 - \frac{p_0}{p_1}} \log \frac{p_1}{p_0} - 1 \right] = 1 + \mathcal{D}_\phi(p_0, p_1) \ . \tag{102}$$

which we rewrote using an f-divergence defined by $\phi(x) = \frac{-\log(x)}{1-x} - 1$. Similarly, we obtain

$$J_2 = \int_{\mathbb{R}^d} \int_0^1 \frac{p_0(\boldsymbol{x})^2}{(1-t)p_0(\boldsymbol{x}) + tp_1(\boldsymbol{x})} dtd\boldsymbol{x} = \int_{\mathbb{R}^d} \frac{p_0(\boldsymbol{x})^2}{p_1(\boldsymbol{x})} \left( \int_0^1 \frac{1}{\frac{p_0(\boldsymbol{x})}{p_1(\boldsymbol{x})} + t(1 - \frac{p_0(\boldsymbol{x})}{p_1(\boldsymbol{x})})} dt \right) d\boldsymbol{x} \tag{103}$$

$$= \int_{\mathbb{R}^d} \frac{p_0(\boldsymbol{x})^2}{p_1(\boldsymbol{x})} \left( \frac{1}{\frac{p_0}{p_1} - 1} \log \frac{p_0}{p_1} \right) d\boldsymbol{x} = 1 + \mathbb{E}_{p_0} \left[ \frac{1}{1 - \frac{p_1}{p_0}} \log \frac{p_0}{p_1} - 1 \right] = 1 + \mathcal{D}_\phi(p_1, p_0) \ . \tag{104}$$

Putting this together, we get

$$\frac{1}{N} \int_0^1 I(t)dt \le \frac{1}{N}(2 + \mathcal{D}_\phi(p_0, p_1) + \mathcal{D}_\phi(p_1, p_0)) \ . \tag{105}$$

How does this divergence depend on the parameter-distance $\|\boldsymbol{\theta}_1 - \boldsymbol{\theta}_0\|$? Does it bring down the dependency from exponential to something lower? We next analyze this:

$$\mathcal{D}_\phi(p_0, p_1) + 1 = \mathbb{E}_{p_1} \frac{1}{1 - \frac{p_0}{p_1}} \log \frac{p_1}{p_0}$$

which looks like a Kullback-Leibler divergence, where the integrand is reweighted by $\frac{1}{1 - \frac{p_0(\boldsymbol{x})}{p_1(\boldsymbol{x})}}$. Note that

$$\begin{cases} \frac{1}{1 - \frac{p_0(\boldsymbol{x})}{p_1(\boldsymbol{x})}} \ge 1 & p_0(\boldsymbol{x}) \le p_1(\boldsymbol{x}) \\ \frac{1}{1 - \frac{p_0(\boldsymbol{x})}{p_1(\boldsymbol{x})}} < 1 & p_0(\boldsymbol{x}) > p_1(\boldsymbol{x}) \end{cases} \tag{106}$$

which motivates separating the integral over both domains

$$1 + \mathcal{D}_\phi(p_0, p_1) = \int_{\{\boldsymbol{x} \in \mathbb{R}^D | p_0(\boldsymbol{x}) \le p_1(\boldsymbol{x})\}} p_1(\boldsymbol{x}) \log \frac{p_1(\boldsymbol{x})}{p_0(\boldsymbol{x})} \frac{1}{1 - \frac{p_0(\boldsymbol{x})}{p_1(\boldsymbol{x})}} \tag{107}$$

$$+ \int_{\{\boldsymbol{x} \in \mathbb{R}^D | p_0(\boldsymbol{x}) > p_1(\boldsymbol{x})\}} p_1(\boldsymbol{x}) \log \frac{p_1(\boldsymbol{x})}{p_0(\boldsymbol{x})} \frac{1}{1 - \frac{p_0(\boldsymbol{x})}{p_1(\boldsymbol{x})}} \tag{108}$$

$$\le \int_{\{\boldsymbol{x} \in \mathbb{R}^D | p_0(\boldsymbol{x}) \le p_1(\boldsymbol{x})\}} p_1(\boldsymbol{x}) + \int_{\{\boldsymbol{x} \in \mathbb{R}^D | p_0(\boldsymbol{x}) > p_1(\boldsymbol{x})\}} p_1(\boldsymbol{x}) \log \frac{p_1(\boldsymbol{x})}{p_0(\boldsymbol{x})} \tag{109}$$

$$\le 1 + D_{\mathrm{KL}}(p_1, p_0) \tag{110}$$

Hence we get

$$\frac{1}{N} \int_0^1 I(t)dt \leq \frac{1}{N} \times (2 + \mathcal{D}_{\mathrm{KL}}(p_0, p_1) + \mathcal{D}_{\mathrm{KL}}(p_1, p_0)) \ . \tag{111}$$

We now suppose the proposal and target are distributions in an exponential family. The KL divergence between exponential distributions with parameters $\boldsymbol{\theta}_0$ and $\boldsymbol{\theta}_1$, is given by the Bregman divergence of the log-partition on the swapped parameters [34, Eq. 29]

$$\mathcal{D}_{\mathrm{KL}}(p_0, p_1) = \mathcal{D}_{\log Z}^{\mathrm{Bregman}}(\boldsymbol{\theta}_1, \boldsymbol{\theta}_0) := \log Z(\boldsymbol{\theta}_1) - \log Z(\boldsymbol{\theta}_0) - \nabla \log Z(\boldsymbol{\theta}_0)(\boldsymbol{\theta}_1 - \boldsymbol{\theta}_0) \tag{112}$$

$$\leq \frac{L}{2} \|\boldsymbol{\theta}_1 - \boldsymbol{\theta}_0\|^2 \tag{113}$$

Hence

$$\mathrm{MSE} \leq \frac{1}{N} \times (2 + L\|\|\boldsymbol{\theta}_1 - \boldsymbol{\theta}_0\|^2) + o\left(\frac{1}{N}\right) + o\left(\frac{K^2}{N}\right) \tag{114}$$

using the $L$-smoothness of the log-partition function $\log Z(\boldsymbol{\theta})$.

**Discussion on the assumptions for theorems 2, 3, 4, 5** For these theorems, we have supposed that the target and proposal distributions are in an exponential family with a log partition that verifies

$$M \, \mathbf{Id} \preccurlyeq \nabla_{\boldsymbol{\theta}}^2 \log Z(\boldsymbol{\theta}) \preccurlyeq L \, \mathbf{Id} \ . \tag{115}$$

We now look at the validity of this assumption for a simple example: the univariate Gaussian, which is in an exponential family. The canonical parameters are its mean and variance $(\mu, v)$. Written as an exponential family,

$$p(\boldsymbol{x}) := \exp(\langle \boldsymbol{\theta}, \boldsymbol{t}(\boldsymbol{x}) \rangle - \log Z(\boldsymbol{\theta})) \tag{116}$$

the natural parameters are $\boldsymbol{\theta} = (\mu/v, -1/(2v))$, associated with the sufficient statistics $\boldsymbol{t}(x) = (x, x^2)$ [34]. The log-partition function and its derivatives are

$$\log Z(\boldsymbol{\theta}) = -\frac{\theta_1^2}{4\theta_2} - \frac{1}{2} \log(-2\theta_2) \tag{117}$$

$$\nabla \log Z(\boldsymbol{\theta}) = \mathbb{E}_{x \sim p}[t(x)] = \left(-\frac{\theta_1}{2\theta_2}, -\frac{1}{2\theta_2} + \frac{\theta_1^2}{4\theta_2^2}\right) \tag{118}$$

$$\nabla^2 \log Z(\boldsymbol{\theta}) = \mathrm{Var}_{x \sim p}[t(x)] = \frac{1}{2\theta_2} \begin{pmatrix} -1 & \frac{\theta_1}{\theta_2} \\ \frac{\theta_1}{\theta_2} & \frac{1}{\theta_2} - \frac{1}{2}\frac{\theta_1^2}{\theta_2} \end{pmatrix} = \begin{pmatrix} v & 2\mu v \\ 2\mu v & 2v^2 - \mu^2 \end{pmatrix} \tag{119}$$

When the mean is zero, the eigenvalues of the hessian are in fact the diagonal values $(v, 2v^2)$, and they are bounded if and only if the variance $v$ is bounded.

**Proof of Theorem 6** *Constant error of annealed NCE with the arithmetic path and "oracle-trig" schedule*

We now study the estimation error produced by the arithmetic path with "oracle-trig" schedule (table 2, line 5). We write the optimal path of Eq. 66 in the limit when the distributions have little overlap, pointwise (with error $\epsilon'$) and on average (with error $\epsilon$):

$$\sqrt{p_0(\boldsymbol{x})p_1(\boldsymbol{x})} = \epsilon'(\boldsymbol{x}) \tag{120}$$

$$\int \sqrt{p_0(\boldsymbol{x})p_1(\boldsymbol{x})}d\boldsymbol{x} = \epsilon \tag{121}$$

We briefly note that there exist certain conditions, given by the dominated convergence theorem, where the first error (pointwise) going to zero implies the second (on average) going to zero as well, but this is outside the scope of this proof. Now, many relevant quantities involved in the optimal distribution and its estimation error simplify as $\epsilon'(\boldsymbol{x}) \to 0$ pointwise and $\epsilon \to 0$. The notation $O(\cdot)$ will hide dependencies on absolute constants only.

$$\mathcal{D}_{H^2}(p_0, p_1) := 1 - \int \sqrt{p_0 p_1} = 1 - \epsilon \tag{122}$$

$$\alpha_H := \arctan\left(\sqrt{\frac{\mathcal{D}_{H^2}(p_0, p_1)}{2 - \mathcal{D}_{H^2}(p_0, p_1)}}\right) = \frac{\pi}{4} - \frac{\epsilon}{2} + o(\epsilon) \tag{123}$$

$$a_t := \frac{\cos((2t-1)\alpha_H)}{2\cos(\alpha_H)} - \frac{\sin((2t-1)\alpha_H)}{2\sin(\alpha_H)} = \cos\left(\frac{\pi t}{2}\right) + \epsilon(t-1)\sin\left(\frac{\pi t}{2}\right) + o(\epsilon) \tag{124}$$

$$b_t = \frac{\cos((2t-1)\alpha_H)}{2\cos(\alpha_H)} + \frac{\sin((2t-1)\alpha_H)}{2\sin(\alpha_H)} = \sin\left(\frac{\pi t}{2}\right) - \epsilon t\cos\left(\frac{\pi t}{2}\right) + o(\epsilon) \tag{125}$$

$$\partial_t a_t := -\alpha_H\left(\frac{\sin((2t-1)\alpha_H)}{\cos(\alpha_H)} + \frac{\cos((2t-1)\alpha_H)}{\sin(\alpha_H)}\right) \tag{126}$$

$$= -\frac{\pi}{2}\sin\left(\frac{\pi t}{2}\right) + \epsilon\left(\sin\left(\frac{\pi t}{2}\right) + \frac{\pi}{2}(t-1)\cos\left(\frac{\pi t}{2}\right)\right) + o(\epsilon) \tag{127}$$

$$\partial_t b_t := -\alpha_H\left(\frac{\sin((2t-1)\alpha_H)}{\cos(\alpha_H)} - \frac{\cos((2t-1)\alpha_H)}{\sin(\alpha_H)}\right) \tag{128}$$

$$= \frac{\pi}{2}\cos\left(\frac{\pi t}{2}\right) + \epsilon\left(\frac{\pi}{2}t\sin\left(\frac{\pi t}{2}\right) - \cos\left(\frac{\pi t}{2}\right)\right) + o(\epsilon) \tag{129}$$

$$a_t \times \partial_t a_t := -\frac{\pi}{4}\sin(\pi t) + \epsilon\left(\frac{1}{2}\sin(\pi t) + \frac{\pi}{2}(t-1)\cos(\pi t)\right) + o(\epsilon) \tag{130}$$

$$a_t \times \partial_t b_t := \frac{\pi}{2}\cos^2\left(\frac{\pi t}{2}\right) + \epsilon\left(\frac{\pi}{2}(1-2t)\sin(\pi t) + \cos(\pi t) + 1\right) + o(\epsilon) \tag{131}$$

$$b_t \times \partial_t a_t := -\frac{\pi}{2}\sin^2\left(\frac{\pi t}{2}\right) + \epsilon\left(\frac{1}{2} - \frac{1}{2}\cos(\pi t) + \frac{1}{2}\frac{\pi}{2}\sin(\pi t)(2t-1)\right) + o(\epsilon) \tag{132}$$

$$b_t \times \partial_t b_t := \frac{\pi}{4}\sin(\pi t) + \epsilon\left(-\frac{1}{2}\sin(\pi t) - \frac{\pi}{2}t\cos(\pi t)\right) + o(\epsilon) \tag{133}$$

This leads to the following simplification of the optimal path

$$p_t^{\text{opt}}(\boldsymbol{x}) := \left(a_t\sqrt{p_0(\boldsymbol{x})} + b_t\sqrt{p_1(\boldsymbol{x})}\right)^2 = a(t)^2 p_0(\boldsymbol{x}) + b(t)^2 p_1(\boldsymbol{x}) + 2a(t)b(t)\epsilon'(\boldsymbol{x}) \tag{134}$$

$$= a(t)^2 p_0(\boldsymbol{x}) + b(t)^2 p_1(\boldsymbol{x}) + 2a(t)b(t)\epsilon'(\boldsymbol{x}) \tag{135}$$

$$= \cos^2\left(\frac{\pi t}{2}\right)p_0(\boldsymbol{x}) + \sin^2\left(\frac{\pi t}{2}\right)p_1(\boldsymbol{x}) + \epsilon'(\boldsymbol{x})\sin(\pi t) \tag{136}$$

$$+ \epsilon\sin(\pi t)(p_0(\boldsymbol{x})(t-1) - p_1(\boldsymbol{x})t) + \epsilon\epsilon'(\boldsymbol{x})((1-2t)\cos(\pi t) - 1) + o(\epsilon) \tag{137}$$

$$= p_t^{\text{arith-trig}}(\boldsymbol{x}) + O(\epsilon'(\boldsymbol{x})) + O(\epsilon g_1(\boldsymbol{x})) \tag{138}$$

where we denoted by

$$p_t^{\text{arith-trig}}(\boldsymbol{x}) := \cos^2\left(\frac{\pi t}{2}\right)p_0(\boldsymbol{x}) + \sin^2\left(\frac{\pi t}{2}\right)p_1(\boldsymbol{x}) \tag{139}$$

the arithmetic path with "oracle-trig" schedule defined in table 2 (line 5); the trigonometric weights in evolve slowly at the end points $t = 0$ and $t = 1$. We also define $g_1(\boldsymbol{x}) = \sin(\pi t)(p_0(\boldsymbol{x})(t-1) - p_1(\boldsymbol{x})t)$ which is an integrable function. This proves the first part of this theorem, which is that the optimal path $p^{\text{opt}}$ is close to a certain arithmetic path (with trigonometric weights) $p^{\text{arith-trig}}$, and that closeness is controlled by how little overlap there is between the endpoint distributions $p_0$ and $p_1$, on average and pointwise.

Similarly, we can control how close these two paths are in terms of estimation errors. The estimation error in Eq. 70 produced by a path is

$$\text{MSE} := \frac{1}{N}\int_0^1 I(t)dt + o\left(\frac{1}{N}\right) + o\left(\frac{K^2}{N}\right) \tag{140}$$

$$= \frac{1}{N}\int_{\mathbb{R}^D}\int_0^1 \left(\partial_t \log p_t(\boldsymbol{x})\right)^2 p_t(\boldsymbol{x})dtd\boldsymbol{x} + o\left(\frac{1}{N}\right) + o\left(\frac{K^2}{N}\right) \tag{141}$$

$$= \frac{1}{N}\int_{\mathbb{R}^D}\int_0^1 \left(\frac{\partial_t p_t(\boldsymbol{x})}{p_t(\boldsymbol{x})}\right)^2 p_t(\boldsymbol{x})dtd\boldsymbol{x} + o\left(\frac{1}{N}\right) + o\left(\frac{K^2}{N}\right) \tag{142}$$

We denote by $\text{MSE}_{\text{optimal}}$ and $\text{MSE}_{\text{arith-trig}}$ the estimation errors produced respectively by the optimal path and the arithmetic path with trigonometric weights. The estimation error of the optimal path can be Taylor-expanded in terms of the overlap ($\epsilon$ and $\epsilon'(\boldsymbol{x})$) as well. We next compute the intermediate quantities that are required to obtain the estimation error.

$$\partial_t p_t^{\text{opt}}(\boldsymbol{x}) := \partial_t\left(\left(a_t\sqrt{p_0(\boldsymbol{x})} + b_t\sqrt{p_1(\boldsymbol{x})}\right)^2\right) \tag{143}$$

$$= 2(\partial_t a_t\sqrt{p_0(\boldsymbol{x})} + \partial_t b_t\sqrt{p_1(\boldsymbol{x})})(a_t\sqrt{p_0(\boldsymbol{x})} + b_t\sqrt{p_1(\boldsymbol{x})}) \tag{144}$$

$$= 2p_0(\boldsymbol{x})\left(a_t \times \partial_t a_t\right) + 2p_1(\boldsymbol{x})\left(b_t \times \partial_t b_t\right) + \tag{145}$$

$$2\sqrt{p_0(\boldsymbol{x})p_1(\boldsymbol{x})}\left(\partial_t a_t \times b_t + a_t \times \partial_t b_t\right) \tag{146}$$

$$= \pi\sin(\pi t)\left(\frac{p_1(\boldsymbol{x}) - p_0(\boldsymbol{x})}{2}\right) + \pi\cos(\pi t)\sqrt{p_0(\boldsymbol{x})p_1(\boldsymbol{x})} + \epsilon\Big( \tag{147}$$

$$p_0(\boldsymbol{x})\left(\sin(\pi t) + \pi(t-1)\cos(\pi t)\right) + p_1(\boldsymbol{x})\left(-\sin(\pi t) - \pi t\cos(\pi t)\right) + \tag{148}$$

$$\sqrt{p_0(\boldsymbol{x})p_1(\boldsymbol{x})}\left(\sin(\pi t)(1 - 2t)\frac{\pi}{2} + \cos(\pi t) + 3\right)\Big) + o(\epsilon) \tag{149}$$

$$= \pi\sin(\pi t)\left(\frac{p_1(\boldsymbol{x}) - p_0(\boldsymbol{x})}{2}\right) + O(\epsilon'(\boldsymbol{x})) + O(\epsilon g_2(\boldsymbol{x})) \tag{150}$$

$$= \partial_t p_t^{\text{arith-trig}}(\boldsymbol{x}) + O(\epsilon'(\boldsymbol{x})) + O(\epsilon g_2(\boldsymbol{x})) \tag{151}$$

where $g_2(\boldsymbol{x}) = p_0(\boldsymbol{x})\left(\sin(\pi t) + \pi(t-1)\cos(\pi t)\right) + p_1(\boldsymbol{x})\left(-\sin(\pi t) - \pi t\cos(\pi t)\right)$ is an integrable function. It follows,

$$\left(\partial_t p_t(\boldsymbol{x})\right)^2 = \left(\partial_t p_t^{\text{arith-trig}}(\boldsymbol{x}) + O(\epsilon'(\boldsymbol{x})) + O(\epsilon g_2(\boldsymbol{x}))\right)^2 \tag{152}$$

$$= \left(\partial_t p_t^{\text{arith-trig}}(\boldsymbol{x})\right)^2 + O(\epsilon'(\boldsymbol{x})) + O(\epsilon g_2(\boldsymbol{x})) \tag{153}$$

by expanding and using that $\partial_t p_t^{\text{arith-trig}}(\boldsymbol{x}) = \pi\sin(\pi t)\left(\frac{p_1(\boldsymbol{x}) - p_0(\boldsymbol{x})}{2}\right)$ is bounded so that $\partial_t p_t^{\text{arith-trig}}(\boldsymbol{x}) \times \epsilon'(\boldsymbol{x}) = O(\epsilon'(\boldsymbol{x}))$. Moreover,

$$\frac{1}{p_t^{\text{opt}}(\boldsymbol{x})} = \frac{1}{p_t^{\text{arith-trig}}(\boldsymbol{x}) + O(\epsilon'(\boldsymbol{x})) + O(\epsilon g_1(\boldsymbol{x}))} = \frac{1}{p_t^{\text{arith-trig}}(\boldsymbol{x})}\frac{1}{1 + \frac{O(\epsilon'(\boldsymbol{x})) + O(\epsilon g_1(\boldsymbol{x}))}{p_t^{\text{arith-trig}}(\boldsymbol{x})}}.$$
$$\tag{154}$$

Denoting by $\epsilon''(\boldsymbol{x}) := (O(\epsilon'(\boldsymbol{x})) + O(\epsilon g_1(\boldsymbol{x})))/p_t^{\text{arith-trig}}(\boldsymbol{x})$ a third quantity which we assume goes to zero, we get

$$\frac{1}{p_t^{\text{opt}}(\boldsymbol{x})} = \frac{1}{p_t^{\text{arith-trig}}(\boldsymbol{x})} + \frac{O(\epsilon''(\boldsymbol{x}))}{p_t^{\text{arith-trig}}(\boldsymbol{x})} \quad . \tag{155}$$

We can now write

$$(\partial_t \log p_t(\boldsymbol{x}))^2 \times p_t(\boldsymbol{x}) = \frac{(\partial_t p_t(\boldsymbol{x}))^2}{p_t(\boldsymbol{x})} \tag{156}$$

$$= \left( (\partial_t p_t^{\text{arith-trig}}(\boldsymbol{x}))^2 + O(\epsilon'(\boldsymbol{x})) + O(\epsilon g_2(\boldsymbol{x})) \right) \times \left( \frac{1}{p_t^{\text{arith-trig}}(\boldsymbol{x})} + \frac{O(\epsilon''(\boldsymbol{x}))}{p_t^{\text{arith-trig}}(\boldsymbol{x})} \right) \tag{157}$$

$$= \frac{(\partial_t p_t^{\text{arith-trig}}(\boldsymbol{x}))^2}{p_t^{\text{arith-trig}}(\boldsymbol{x})} + \frac{O(\epsilon'(\boldsymbol{x})) + O(\epsilon g_2(\boldsymbol{x}))}{p_t^{\text{arith-trig}}(\boldsymbol{x})} + \frac{(\partial_t p_t^{\text{arith-trig}}(\boldsymbol{x}))^2}{p_t^{\text{arith-trig}}(\boldsymbol{x})} O(\epsilon''(\boldsymbol{x})) \tag{158}$$

$$= \frac{(\partial_t p_t^{\text{arith-trig}}(\boldsymbol{x}))^2}{p_t^{\text{arith-trig}}(\boldsymbol{x})} + O(\epsilon''(\boldsymbol{x})) \left( 1 + \frac{(\partial_t p_t^{\text{arith-trig}}(\boldsymbol{x}))^2}{p_t^{\text{arith-trig}}(\boldsymbol{x})} \right) \tag{159}$$

by expanding. We can then write

$$\text{MSE}_{\text{optimal}} \tag{160}$$

$$= \frac{1}{N} \int_{\mathbb{R}^D} \int_0^1 \frac{(\partial_t p_t^{\text{opt}}(\boldsymbol{x}))^2}{p_t^{\text{opt}}(\boldsymbol{x})} dt d\boldsymbol{x} + o\left(\frac{1}{N}\right) + o\left(\frac{K^2}{N}\right) \tag{161}$$

$$= \frac{1}{N} \int_{\mathbb{R}^D} \int_0^1 \left( \frac{(\partial_t p_t^{\text{arith-trig}}(\boldsymbol{x}))^2}{p_t^{\text{arith-trig}}(\boldsymbol{x})} + O(\epsilon''(\boldsymbol{x})) \left( 1 + \frac{(\partial_t p_t^{\text{arith-trig}}(\boldsymbol{x}))^2}{p_t^{\text{arith-trig}}(\boldsymbol{x})} \right) \right) dt d\boldsymbol{x} \tag{162}$$

$$+ o\left(\frac{1}{N}\right) + o\left(\frac{K^2}{N}\right) \tag{163}$$

$$= \text{MSE}_{\text{arith-trig}} + \frac{1}{N} \int_{\mathbb{R}^D} \int_0^1 O(\epsilon''(\boldsymbol{x})) \left( 1 + \frac{(\partial_t p_t^{\text{arith-trig}}(\boldsymbol{x}))^2}{p_t^{\text{arith-trig}}(\boldsymbol{x})} \right) dt d\boldsymbol{x} + o\left(\frac{1}{N}\right) + o\left(\frac{K^2}{N}\right) \tag{164}$$

$$= \text{MSE}_{\text{arith-trig}} + O\left(\frac{\epsilon'''}{N}\right) + o\left(\frac{1}{N}\right) + o\left(\frac{K^2}{N}\right) \tag{165}$$

Additionally, we assume that the remainder term denoted by

$$\epsilon''' = \int_{\mathbb{R}^D} \int_0^1 O(\epsilon''(\boldsymbol{x})) \left( 1 + \frac{(\partial_t p_t^{\text{arith-trig}}(\boldsymbol{x}))^2}{p_t^{\text{arith-trig}}(\boldsymbol{x})} \right) dt d\boldsymbol{x} \tag{166}$$

is integrable and also goes to zero. On the other hand, the estimation error produced by the optimal path is known [17, Eq. 48] and the result can be Taylor-expanded as well

$$\text{MSE}_{\text{optimal}} = \frac{1}{N} 16 \alpha_H^2 + o\left(\frac{1}{N}\right) + o\left(\frac{K^2}{N}\right) \tag{167}$$

$$= \frac{1}{N} 16 \left( \frac{\pi^2}{16} + O(\epsilon) \right) + o\left(\frac{1}{N}\right) + o\left(\frac{K^2}{N}\right) \tag{168}$$

$$= \frac{1}{N} \pi^2 + O\left(\frac{\epsilon}{N}\right) + o\left(\frac{1}{N}\right) + o\left(\frac{K^2}{N}\right) \quad . \tag{169}$$

This means that

$$\text{MSE}_{\text{arith-trig}} = \frac{1}{N} \pi^2 + O\left(\frac{\epsilon}{N}\right) + o\left(\frac{1}{N}\right) + o\left(\frac{K^2}{N}\right) + O\left(\frac{\epsilon'''}{N}\right) \quad . \tag{170}$$

## C  Useful Lemma

**Lemma 1** *Chi-square divergence of between a density and a mixture We wish to upper bound the chi-square divergence between a distribution $p(\boldsymbol{x})$ and a mixture $wp(\boldsymbol{x}) + (1-w)q(\boldsymbol{x})$, where $0 < w < 1$.*

$$\mathcal{D}_{\chi^2}(p, wp + (1-w)q) = \int_{\boldsymbol{x} \in \mathbb{R}^D} \frac{p(\boldsymbol{x})^2}{wp(\boldsymbol{x}) + (1-w)q(\boldsymbol{x})} d\boldsymbol{x} - 1 \tag{171}$$

$$= \int_{\{\boldsymbol{x} \in \mathbb{R}^D | p(\boldsymbol{x}) < q(\boldsymbol{x})\}} \frac{p(\boldsymbol{x})^2}{wp(\boldsymbol{x}) + (1-w)q(\boldsymbol{x})} d\boldsymbol{x} \tag{172}$$

$$+ \int_{\{\boldsymbol{x} \in \mathbb{R}^D | p(\boldsymbol{x}) > q(\boldsymbol{x})\}} \frac{p(\boldsymbol{x})^2}{wp(\boldsymbol{x}) + (1-w)q(\boldsymbol{x})} d\boldsymbol{x} - 1 \tag{173}$$

$$\leq \int_{\{\boldsymbol{x} \in \mathbb{R}^D | p(\boldsymbol{x}) < q(\boldsymbol{x})\}} \frac{p(\boldsymbol{x})^2}{wp(\boldsymbol{x}) + (1-w)p(\boldsymbol{x})} d\boldsymbol{x} \tag{174}$$

$$+ \int_{\{\boldsymbol{x} \in \mathbb{R}^D | p(\boldsymbol{x}) > q(\boldsymbol{x})\}} \frac{p(\boldsymbol{x})^2}{wq(\boldsymbol{x}) + (1-w)q(\boldsymbol{x})} d\boldsymbol{x} - 1 \tag{175}$$

$$\leq \int_{\boldsymbol{x} \in \mathbb{R}^D} p(\boldsymbol{x}) d\boldsymbol{x} + \int_{\boldsymbol{x} \in \mathbb{R}^D} \frac{p(\boldsymbol{x})^2}{q(\boldsymbol{x})} d\boldsymbol{x} - 1 \tag{176}$$

$$= \int_{\boldsymbol{x} \in \mathbb{R}^D} \frac{p(\boldsymbol{x})^2}{q(\boldsymbol{x})} d\boldsymbol{x} = \mathcal{D}_{\chi^2}(p, q) + 1 \tag{177}$$

