*where $f$ is an increasing function defined in Appendix B.3.*

| Path name | Unnormalized density | Normalized density | Error |
|---|---|---|---|
| Geometric | $f_t(\boldsymbol{x}) = p_0(\boldsymbol{x})^{1-t} f_1(\boldsymbol{x})^t$ | $p_t(\boldsymbol{x}) \propto p_0(\boldsymbol{x})^{1-t} p_1(\boldsymbol{x})^t$ | poly |
| Arithmetic | $f_t(\boldsymbol{x}) = (1 - w_t) p_0(\boldsymbol{x}) + w_t f_1(\boldsymbol{x})$ | $p_t(\boldsymbol{x}) = (1 - \tilde{w}_t) p_0(\boldsymbol{x}) + \tilde{w}_t p_1(\boldsymbol{x})$ | |
| vanilla | $w_t = t$ | $\tilde{w}_t = \frac{t Z_1}{(1-t) + t Z_1}$ | exp |
| oracle | $w_t = \frac{t}{t + Z_1(1-t)}$ | $\tilde{w}_t = t$ | poly |
| oracle-trig | $w_t = \frac{\sin^2\left(\frac{\pi t}{2}\right)}{\sin^2\left(\frac{\pi t}{2}\right) + Z_1 \cos^2\left(\frac{\pi t}{2}\right)}$ | $\tilde{w}_t = \sin^2\left(\frac{\pi t}{2}\right)$ | const |

Table 2: Geometric and arithmetic paths, defined in the space of unnormalized densities (second column); "oracle" and "oracle-trig" are reparameterizations of the arithmetic path which depend on the true normalization $Z_1$. The corresponding normalized densities (third column) produce an estimation error (fourth column) which we quantify.

We suggest an intuitive explanation for this negative result. We begin with the observation that the estimation error (Eq. 8) depends on the *normalized* path of densities. Suppose the target model is rescaled by a constant 100, so that the new unnormalized target density is $f_1(\boldsymbol{x}) \times 100$ and its new normalization is $Z_1 \times 100$. Looking at table 2, this rescaling does not modify the geometric path of normalized densities, while it does the arithmetic path of normalized densities. Because the estimation error depends on path of normalized densities, this makes the arithmetic choice sensitive to target normalization, even more so as the parameter distance grows and the log-normalization with it, as a strongly convex function of it (Appendix, Eq. 85). This suggests making the arithmetic path of normalized distributions "robust" to the choice of $Z_1$. We will show this can be achieved by re-parameterizing the path in terms of $Z_1$.

We next prove that certain reparameterizations can bring down the error to a polynomial and even constant function of the parameter-distance between the target and proposal. The following theorems may seem purely theoretical, as if necessitating an oracle for $Z_1$, but they will actually lead to an efficient estimation algorithm later.

**Theorem 5** (Polynomial error of annealed NCE with an arithmetic path and oracle) *Assume the same as in Theorem 4, replacing the strong convexity of the log-partition by smoothness (Eq. 12). Additionally, suppose an oracle gives the normalization constant $Z_1$ to be used only in the reparameterization of the arithmetic path with $t \to \frac{t}{t + Z_1(1-t)}$ (see Table 2). This brings down the estimation error of annealed NCE to (at most) polynomial in the parameter-distance*

$$\mathrm{MSE}_{\mathrm{NCE}}(p_0, p_1; q, K, N) \leq \frac{1}{N}(2 + L\|\boldsymbol{\theta}_1 - \boldsymbol{\theta}_0\|^2), \qquad \text{when } K \to \infty, q = 1 \qquad (16)$$

*where $L$ is the smoothness constant of $\log Z(\boldsymbol{\theta})$.*

In fact, supposing we have (oracle) access to the normalizing constant $Z_1$, the arithmetic path can even be reparameterized such that it is the optimal path in a certain limit. We next prove such optimality in the limits of a continuous path $K \to \infty$ and "far-away" target and proposal:

**Theorem 6** (Constant error of annealed NCE with an arithmetic path and oracle) *Suppose we can successively take the limit of a continuous annealing path $K \to \infty$, then the limit of a target distribution that tends toward no overlap with the proposal $p_1(\boldsymbol{x}) p_0(\boldsymbol{x}) \to 0$ pointwise (and assuming domination by an integrable function)* [3]. *Then the optimal annealing path convergences pointwise to an arithmetic path reparameterized trigonometrically with $t \to \frac{t}{\sin^2(\frac{\pi t}{2}) + Z_1(1 - \sin^2(\frac{\pi t}{2}))}$. In that limit, the estimation error is constant with respect to the parameter-distance*

$$\mathrm{MSE}_{\mathrm{NCE}}(p_0, p_1; q, K, N) \sim \frac{2\pi^2}{N}, \qquad \text{when } K \to \infty, q = 1 \qquad (17)$$

**Two-step estimation** Thus, we see that, perhaps unsurprisingly, the "optimal" mixture weights in the space of unnormalized densities depends on the true $Z_1$: however, this dependency is simple. We

---

[3]this effectively assumes that $K \to \infty$ faster than $p_1(\boldsymbol{x}) p_0(\boldsymbol{x}) \to 0$, so that the error in the first limit is dominated by the error in the second limit.

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

, it is better to use IS due to its computational simplicity (since the statistical efficiency is the same as NCE). Annealing can always provide substantial benefits (Theorem 2). Moreover, if we have a reasonable a priori estimate of $Z_1$, the arithmetic path achieves very low error (Theorem 5) — sometimes even approaching optimal (Theorem 6). On the other hand, even absent an initial estimate of $Z_1$, the geometric path can exponentially reduce the estimation error compared with no annealing (Theorems 2 and 3).

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

## A  No annealing, $K = 1$

We use [22, Eq.21] for the estimation error of any suitably parameterized [4] classifier $F(\boldsymbol{x}; \boldsymbol{\beta})$ between two distributions $p_1$ and $p_0$. The estimation error is measured by the Mean-Squared Error (MSE)

$$\mathrm{MSE}_{\hat{\boldsymbol{\beta}}}(p_n, \nu, \phi, N) = \frac{\nu + 1}{N} \mathrm{tr}(\boldsymbol{\Sigma}) \tag{18}$$

which depends on the sample sizes $N = N_1 + N_0$, their ratio $\nu = N_1/N_0$, the Bregman classification loss indexed by the convex function $\phi(x)$, and the asymptotic variance matrix

$$\boldsymbol{\Sigma} = \boldsymbol{I}_w^{-1} \big( \boldsymbol{I}_v - (1 + \frac{1}{\nu}) \boldsymbol{m}_w \boldsymbol{m}_w^\top \big) \boldsymbol{I}_w^{-1} \ . \tag{19}$$

Here, $\boldsymbol{m}_w(\boldsymbol{\beta}^*)$, $\boldsymbol{I}_w(\boldsymbol{\beta}^*)$ and $\boldsymbol{I}_v(\boldsymbol{\beta}^*)$ are the reweighted mean and covariances of the paramete-gradient of the classifier, also known as the "relative" Fisher score $\nabla_{\boldsymbol{\beta}} F(\boldsymbol{x}; \beta^*)$,

$$\boldsymbol{m}_w(\boldsymbol{\beta}^*) = \mathbb{E}_{\boldsymbol{x} \sim p_d} \big[ w(\boldsymbol{x}) \nabla_{\boldsymbol{\beta}} F(\boldsymbol{x}; \boldsymbol{\beta}^*) \big] \tag{20}$$

$$\boldsymbol{I}_w(\boldsymbol{\beta}^*) = \mathbb{E}_{\boldsymbol{x} \sim p_d} \big[ w(\boldsymbol{x}) \nabla_{\boldsymbol{\beta}} F(\boldsymbol{x}; \boldsymbol{\beta}^*) \nabla_{\boldsymbol{\beta}} F(\boldsymbol{x}; \boldsymbol{\beta}^*)^\top \big] \tag{21}$$

$$\boldsymbol{I}_v(\boldsymbol{\beta}^*) = \mathbb{E}_{\boldsymbol{x} \sim p_d} \big[ v(\boldsymbol{x}) \nabla_{\boldsymbol{\beta}} F(\boldsymbol{x}; \boldsymbol{\beta}^*) \nabla_{\boldsymbol{\beta}} F(\boldsymbol{x}; \boldsymbol{\beta}^*)^\top \big] \tag{22}$$

where the reweighting of data points is by $w(\boldsymbol{x}) := \frac{p_1}{\nu p_0}(\boldsymbol{x}) \phi'' \big( \frac{p_1}{\nu p_0}(\boldsymbol{x}) \big)$ and by $v(\boldsymbol{x}) = w(\boldsymbol{x})^2 \frac{\nu p_0(\boldsymbol{x}) + p_1(\boldsymbol{x})}{\nu p_0(\boldsymbol{x})}$, which are all evaluated at the true parameter value $\boldsymbol{\beta}^*$.

**Scalar parameterization**  We now consider a specific parameterization of the classifier:

$$F(\boldsymbol{x}; \beta) = \log \left( \frac{f_1(\boldsymbol{x})}{\nu f_0(\boldsymbol{x})} \right) - \beta \tag{23}$$

where the optimal parameter is the log-ratio of normalizations $\beta^* = \log(Z_1/Z_0)$. Consequently, we have $\nabla_{\boldsymbol{\beta}} F(\boldsymbol{x}; \beta^*) = -1$ and plugging this into the above quantities yields

$$\mathrm{MSE} = \frac{1 + \nu}{T} \left( \frac{\mathbb{E}_{\boldsymbol{x} \sim p_1} \big[ w^2(\boldsymbol{x}) \frac{\nu p_0(\boldsymbol{x}) + p_1(\boldsymbol{x})}{\nu p_0(\boldsymbol{x})} \big]}{\mathbb{E}_{\boldsymbol{x} \sim p_1} [w(\boldsymbol{x})]^2} - \big( 1 + \frac{1}{\nu} \big) \right)$$

which matches the formula found in [20, Eq 3.2]. For different choices of the Bregman classification loss, the estimation error is written using a divergence between the two distributions

| Name | Loss identified by $\phi(\boldsymbol{x})$ | Estimator | MSE |
|---|---|---|---|
| IS | $x \log x$ | $\log \mathbb{E}_{p_0} \frac{f_1}{f_0}$ | $\frac{1+\nu}{\nu N} \mathcal{D}_{\chi^2}(p_1, p_0)$ |
| RevIS | $-\log x$ | $-\log \mathbb{E}_{p_1} \frac{f_0}{f_1}$ | $\frac{1+\nu}{N} \mathcal{D}_{\chi^2}(p_0, p_1)$ |
| NCE | $x \log x - (1 + x) \log(\frac{1+x}{2})$ | implicit | $\frac{(1+\nu)^2}{\nu N} \frac{\mathcal{D}_{\mathrm{HM}}(p_1, p_0)}{1 - \mathcal{D}_{\mathrm{HM}}(p_1, p_0)}$ |
| IS-RevIS | $(1 - \sqrt{x})^2$ | $\log \mathbb{E}_{p_0} \frac{f_1}{f_0} - \log \mathbb{E}_{p_1} \frac{f_0}{f_1}$ | $\frac{(1+\nu)^2}{\nu N} \frac{1 - (1 - \mathcal{D}_{H^2}(p_d, p_n))^2}{(1 - \mathcal{D}_{H^2}(p_d, p_n))^2}$ |

where

$$\mathcal{D}_{\chi^2}(p_1, p_0) := \big( \int \frac{p_1^2}{p_0} \big) - 1 \text{ is the chi-squared divergence}$$

---

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

$$\leq 1 + D_{\text{KL}}(p_1, p_0) \tag{101}$$

Hence we get

$$\text{MSE} \leq \frac{1}{N} \times (2 + \mathcal{D}_{\text{KL}}(p_0, p_1) + \mathcal{D}_{\text{KL}}(p_1, p_0)) \ . \tag{102}$$

We now suppose the proposal and target are distributions in an exponential family. The KL divergence between exponential distributions with parameters $\boldsymbol{\theta}_0$ and $\boldsymbol{\theta}_1$, is given by the Bregman divergence of the log-partition on the swapped parameters [30, Eq. 29]

$$\mathcal{D}_{\text{KL}}(p_0, p_1) = \mathcal{D}_{\log Z}^{\text{Bregman}}(\boldsymbol{\theta}_1, \boldsymbol{\theta}_0) := \log Z(\boldsymbol{\theta}_1) - \log Z(\boldsymbol{\theta}_0) - \nabla \log Z(\boldsymbol{\theta}_0)(\boldsymbol{\theta}_1 - \boldsymbol{\theta}_0) \tag{103}$$

$$\leq \frac{L}{2} \|\boldsymbol{\theta}_1 - \boldsymbol{\theta}_0\|^2 \tag{104}$$

Hence

$$\text{MSE} \leq \frac{1}{N} \times (2 + L\|\|\boldsymbol{\theta}_1 - \boldsymbol{\theta}_0\|^2) \tag{105}$$

using the $L$-smoothness of the log-partition function $\log Z(\boldsymbol{\theta})$.

**Discussion on the assumptions for theorems 2, 3, 4, 5**   For these theorems, we have supposed that the target and proposal distributions are in an exponential family with a log parition that verifies

$$M\,\mathbf{Id} \preccurlyeq \nabla_{\boldsymbol{\theta}}^2 \log Z(\boldsymbol{\theta}) \preccurlyeq L\,\mathbf{Id} \ . \tag{106}$$

We now look at the validity of this assumption for a simple example: the univariate gaussian, which is in an exponential family. The canonical parameters are its mean and variance $(\mu, v)$. Written as an exponential family,

$$p(\boldsymbol{x}) := \exp(\langle \boldsymbol{\theta}, \boldsymbol{t}(\boldsymbol{x}) \rangle - \log Z(\boldsymbol{\theta})) \tag{107}$$

the natural parameters are $\boldsymbol{\theta} = (\mu/v, -1/(2v))$, associated with the sufficient statistics $\boldsymbol{t}(x) = (x, x^2)$ [30]. The log-partition function and its derivatives are

$$\log Z(\boldsymbol{\theta}) = -\frac{\theta_1^2}{4\theta_2} - \frac{1}{2} \log(-2\theta_2) \tag{108}$$

$$\nabla \log Z(\boldsymbol{\theta}) = \mathbb{E}_{x \sim p}[t(x)] = \left( -\frac{\theta_1}{2\theta_2}, -\frac{1}{2\theta_2} + \frac{\theta_1^2}{4\theta_2^2} \right) \tag{109}$$

$$\nabla^2 \log Z(\boldsymbol{\theta}) = \text{Var}_{x \sim p}[t(x)] = \frac{1}{2\theta_2} \begin{pmatrix} -1 & \frac{\theta_1}{\theta_2} \\ \frac{\theta_1}{\theta_2} & \frac{1}{\theta_2} - \frac{1}{2}\frac{\theta_1^2}{\theta_2} \end{pmatrix} = \begin{pmatrix} v & 2\mu v \\ 2\mu v & 2v^2 - \mu^2 \end{pmatrix} \tag{110}$$

609 When the mean is zero, the eigenvalues of the hessian are in fact the diagonal values $(v, 2v^2)$, and
610 they are bounded if and only if the variance $v$ is bounded.

**Proof of Theorem 6** *Constant error of annealed NCE with the arithmetic path and "oracle-trig"*
612 *schedule*

613 We now study the estimation error produced by the arithmetic path with "oracle-trig" schedule
614 (table 2, line 5). We write the optimal path of Eq. 59 in the limit where the distributions do not
615 overlap: $p_0(\boldsymbol{x})p_1(\boldsymbol{x}) \to 0$ pointwise and is bounded by an integrable function. In that limit, many
616 quantities involved in the optimal distribution simplify

$$\mathcal{D}_{H^2}(p_0, p_1) = 1 - \int \sqrt{p_0 p_1} \to 1 \tag{111}$$

$$\alpha_H = \arctan\left(\sqrt{\frac{\mathcal{D}_{H^2}(p_0, p_1)}{2 - \mathcal{D}_{H^2}(p_0, p_1)}}\right) \to \frac{\pi}{4} \tag{112}$$

$$a(t) = \frac{\cos((2t-1)\alpha_H)}{2\cos(\alpha_H)} - \frac{\sin((2t-1)\alpha_H)}{2\sin(\alpha_H)} \to \cos\left(\frac{\pi t}{2}\right) \tag{113}$$

$$b(t) = \frac{\cos((2t-1)\alpha_H)}{2\cos(\alpha_H)} + \frac{\sin((2t-1)\alpha_H)}{2\sin(\alpha_H)} \to \sin\left(\frac{\pi t}{2}\right) . \tag{114}$$

617 All these limits are pointwise: for the first line, the dominated convergence theorem is used to justify
618 the pointwise convergence of the integral $\int p_0 p_1 \to 0$ (L2 convergence of $\sqrt{p_0 p_1}$) and consequently
619 the pointwise convergence of the integral $\int \sqrt{p_0 p_1} \to 0$ (L1 convergence of $\sqrt{p_0 p_1}$). This leads to
620 the following simplification of the optimal path

$$p_t(\boldsymbol{x}) = \left(a(t)\sqrt{p_0(\boldsymbol{x})} + b(t)\sqrt{p_1(\boldsymbol{x})}\right)^2 = a(t)^2 p_0(\boldsymbol{x}) + b(t)^2 p_1(\boldsymbol{x}) + 2a(t)b(t)\sqrt{p_0 p_1} \tag{115}$$

$$\to \cos^2\left(\frac{\pi t}{2}\right)p_0(\boldsymbol{x}) + \sin^2\left(\frac{\pi t}{2}\right)p_1(\boldsymbol{x}) \tag{116}$$

621 which is the arithmetic path with "oracle-trig" schedule defined in table 2 (line 5). The trigonometric
622 weights evolve slowly at the end points $t = 0$ and $t = 1$. The estimation error in Eq. 63 produced by
623 this path converges to

$$\text{MSE} = \frac{1}{N}\int_0^1 I(t)dt = \frac{1}{N}16\alpha_H^2 \sim \frac{1}{N}16\frac{\pi^2}{8} = \frac{1}{N}2\pi^2 . \tag{117}$$

624 which is a constant function of the parameter-distance.

## C  Useful Lemma

**Lemma 1** *Chi-square divergence of between a density and a mixture We wish to upper bound the chi-square divergence between a distribution $p(\boldsymbol{x})$ and a mixture $wp(\boldsymbol{x}) + (1-w)q(\boldsymbol{x})$, where $0 < w < 1$.*

$$\mathcal{D}_{\chi^2}(p, wp + (1-w)q) = \int_{\boldsymbol{x} \in \mathbb{R}^D} \frac{p(\boldsymbol{x})^2}{wp(\boldsymbol{x}) + (1-w)q(\boldsymbol{x})} d\boldsymbol{x} - 1 \tag{118}$$

$$= \int_{\{\boldsymbol{x} \in \mathbb{R}^D \mid p(\boldsymbol{x}) < q(\boldsymbol{x})\}} \frac{p(\boldsymbol{x})^2}{wp(\boldsymbol{x}) + (1-w)q(\boldsymbol{x})} d\boldsymbol{x} \tag{119}$$

$$+ \int_{\{\boldsymbol{x} \in \mathbb{R}^D \mid p(\boldsymbol{x}) > q(\boldsymbol{x})\}} \frac{p(\boldsymbol{x})^2}{wp(\boldsymbol{x}) + (1-w)q(\boldsymbol{x})} d\boldsymbol{x} - 1 \tag{120}$$

$$\leq \int_{\{\boldsymbol{x} \in \mathbb{R}^D \mid p(\boldsymbol{x}) < q(\boldsymbol{x})\}} \frac{p(\boldsymbol{x})^2}{wp(\boldsymbol{x}) + (1-w)p(\boldsymbol{x})} d\boldsymbol{x} \tag{121}$$

$$+ \int_{\{\boldsymbol{x} \in \mathbb{R}^D \mid p(\boldsymbol{x}) > q(\boldsymbol{x})\}} \frac{p(\boldsymbol{x})^2}{wq(\boldsymbol{x}) + (1-w)q(\boldsymbol{x})} d\boldsymbol{x} - 1 \tag{122}$$

$$\leq \int_{\boldsymbol{x} \in \mathbb{R}^D} p(\boldsymbol{x}) d\boldsymbol{x} + \int_{\boldsymbol{x} \in \mathbb{R}^D} \frac{p(\boldsymbol{x})^2}{q(\boldsymbol{x})} d\boldsymbol{x} - 1 \tag{123}$$

$$= \int_{\boldsymbol{x} \in \mathbb{R}^D} \frac{p(\boldsymbol{x})^2}{q(\boldsymbol{x})} d\boldsymbol{x} = \mathcal{D}_{\chi^2}(p, q) + 1 \tag{124}$$