# OpenReview forum: "Provable benefits of annealing for estimating normalizing constants: Importance Sampling, Noise-Contrastive Estimation, and beyond"
_NeurIPS.cc/2023/Conference — NeurIPS 2023 spotlight_

### Official Review · Reviewer_peRm · 2023-06-12

**Soundness:** 3 good
**Presentation:** 3 good
**Contribution:** 3 good
**Rating:** 8
**Confidence:** 3

**Summary:**

The article theoretically bounds the estimation error of NCE depending on the path. The paper first shows that NCE has minimum variance among all estimators using K bridging distributions and then continues by showing that the standard polynomial path can achieve polynomial error in the limit of infinitely many bridging distributions. Then the authors derive first a negative result for the arithmetic path and a positive result on the oracle estimate of the path that uses the unknown normalisation constant. The results are empirically verified.

**Strengths:**

The authors responses clarified the main weaknesses and I am raising my score accordingly.
--------------------------

The paper appears to be well written and some of the results appear to be novel, especially the error bounds. I have not checked the math in the appendix.

The presentation is rather clear and to the point and the experiments nicely validate the theorems.

**Weaknesses:**

- Some related work is missing. importantly, NCE is a method that got discovered and rediscovered several times, see for example this review that calls the same method BAR (Bennets acceptance ratio):

[1] Krause, Oswin, Asja Fischer, and Christian Igel. "Algorithms for estimating the partition function of restricted Boltzmann machines." Artificial Intelligence 278 (2020): 103195.

- As a result of this, Theorem 1 was already known, as [1] reproduces an earlier proof that shows that NCE is a maximum likelihood estimator. Since MLEs are efficient, their variance is bounded by the variance of the Fisher matrix. In that light, the novelty of Theorem 1 is to produce a bound on the Fisher matrix via the path integral. Similarly, [1] already proposed a framework that included the generalisation of several estimators, similar to the Bregman methodology proposed in this work.

- The practical applicability of the results is questionable. While the work shows that the two step estimator works well given perfect samples, it might be very challenging to obtain good samples for the arithmetic path. This is because the modes that appear are disconnected from each other from the very beginning on, which makes it difficult to discover them using MCMC, unlike for example parallel tempering with the polynomial path (see [1]).







**Questions:**

I would suggest to discuss [1] in the article due to its relevance and clear overlap. A few of the references therein might be important as well.

Q1: you stress in the article that your proposed two-step method is not efficient. Do you foresee a way to make it efficient? Is it possible to create a larger experiment on a real estimation task that shows that it can improve on previous estimates?



Finally, on page 5 line 165 where you quote [29], I would propose to also mention Deep Belief networks as an important model class with universal approximation properties for which estimation the normalisation constant is relevant:

Krause, Oswin, et al. "Approximation properties of DBNs with binary hidden units and real-valued visible units." International conference on machine learning. ICML, 2013.



**Limitations:**

yes

---

> ### Author Rebuttal · Authors · 2023-08-08
>
> We thank the reviewer for their valuable feedback and pointing out relevant references. In the following, we will answer the reviewer’s comments point-by-point. We hope we address their concerns and hope the reviewer will consider raising their score.
>
> **“I would suggest to discuss [1] in the article due to its relevance and clear overlap. A few of the references therein might be important as well.”  —** The references [1, 2] suggested by the reviewer are indeed relevant, and we will be sure to add them in the camera-ready version of the paper!
>
> We agree with the reviewer that NCE has been rediscovered under many names, for estimating the normalization constant [3, 4, 5] of an unnormalized model, as pointed out by that reference and some other works [Section 3.2., 6]. Moreover, eq. 5 of [1] uses a similar identity to eq. 1.4 of [7] to generalize importance sampling to a family of estimation methods which includes NCE, also similar to [6].
>
> **“Theorem 1 was already known” —** we agree that a small part of Theorem 1 is known — and we point this out in the paper as well: “In the binary setup [...], the NCE loss is optimal [6, 7] ”. There are several novel parts to Theorem 1 however: (1) we extend the result to a sequence of distributions; (2) we show that the optimality gap between NCE and IS vanishes in the continuous path limit; (3) we provide a closed-form expression of the MSE in that limit.
>
> **“The practical applicability of the results is questionable” —** we agree with the reviewer that bridging the gap between theory and practice is an important challenge; we simply point out that it is an issue for the entire literature. [8] point out that “Due to the difficulty of establishing general results under arbitrary sampling schemes, we shall assume independent draws for theoretical explorations and guidelines”. Similarly, [9] remark that “the [...] analysis assumes perfect transitions which can be unrealistic in practice because many distributions of interest have separated modes between which mixing is difficult.” A number of methods have been developed to deal with sampling issues, including parallel tempering that is mentioned by the reviewer, and tempered transitions. In fact, the path cost for both these methods, parallel tempering [eq. 17, 10] and tempered transitions [eq. 18, 11], are equal (or upper bounded) by a a sum of f-divergences which in the limit of a continuous path is the same cost function as in our Theorem 1. This suggests our results may be applicable to more practical methods in the literature.
>
> **“You stress in the article that your proposed two-step method is not efficient. Do you foresee a way to make it efficient?” —** Our two-step method uses an estimate of the target normalization to reparameterize the arithmetic path. It is not clear how the estimation error of the normalization propagates into an error in the path it reparameterizes. In future work, we hope to look more closely at how our optimality results can be brought to scale.
>
>
> [1] Krause et al. “Algorithms for estimating the partition function of restricted Boltzmann machines.” Artificial Intelligence, 2020.
>
> [2] Krause et al. “Approximation properties of DBNs with binary hidden units and real-valued visible units.” ICML, 2013.
>
> [3] Geyer. “Estimating Normalizing Constants and Reweighting Mixtures”. Technical Report No. 568, School of Statistics University of Minnesota, 1994.
>
> [4] Bennett. “Efficient estimation of free energy differences from Monte Carlo data”. Journal of Computational Physics, 1976.
>
> [5] Gutmann et al. “Noise-Contrastive Estimation of Unnormalized Statistical Models [...]”. Journal of Machine Learning Research, 2012.
>
> [6] Chehab et al., “Optimizing the Noise in Self-Supervised Learning [...]”. Arxiv, 2023.
>
> [7] Meng et al. “Simulating Ratios of Normalizing Constants Via A Simple Identity”. Statistica Sinica, 1996.
>
> [8] Gelman et al. “Simulating normalizing constants: From importance sampling to bridge sampling to path sampling”. Statistical Science, 1998.
>
> [9] Grosse et al. “Annealing between distributions by averaging moments”. NIPS, 2013.
>
> [10] Syed et al. “Parallel tempering on optimized paths”. ICML, 2021.
>
> [11] Behrens et al. “Tuning tempered transitions”. Statistics and Computing, 2012.

---

> > ### Comment · Reviewer_peRm · 2023-08-12
> >
> > Thanks for your reply. I think some of my questions have been answered, but not the most important ones.
> >
> > I still think the authors have still not acknowledged the scope of reference [1] suitably. You write:
> >
> > **we agree that a small part of Theorem 1 is known — and we point this out in the paper as well: “In the binary setup [...], the NCE loss is optimal [6, 7] ”. There are several novel parts to Theorem 1 however: (1) we extend the result to a sequence of distributions;** [...]
> >
> > [1] already includes the derivation of the result beyond a pair of distributions. To be more exact, the authors consider pairs of forward and reverse paths through the set of distributions and then derive NCE for those paths via maximum likelihood.  To the reviewers understandng, this is exactly the setting considered by the article. While points (2) and (3) appear to be novel, this distinction that (1) is not novel should be properly credited.
> >
> > Regarding the practical applicability: i do not see it as an issue that the theoretical work requires perfect samples and I am not assigning any negative points for that. My main point here is that I think that the assumption needs to be reviewed and discussed in terms of its meaning in the two settings. To make it clear: **i believe that the arithmetic path requires a stronger sample oracle and the difference in asymptotic behaviour is a result of that the arithmetic path moves more of the complexity into the sampler**.
> >
> > I am willing to raise my score if these points are discussed in the article.

---

> > > ### Author Response · Authors · 2023-08-12
> > > **Answer to Reviewer peRm**
> > >
> > > We thank the reviewer for pointing out these two points and we will discuss them in the camera-ready version of the paper. We hope this addresses the remaining concerns.
> > >
> > > - **optimality of the NCE estimator for a sequence of distributions**
> > >
> > > We have reread the reference [1] provided by the reviewer: the optimality of the NCE estimator is indeed extended to a sequence of distributions in eq 16 of [1].  **We propose modifying the sentence preceding our Theorem 1 to the following**: “This optimality result has been extended to a sequence of distributions K > 1 [eq. 16, 1]. We show that in the limit of a continuous path, the gap between annealed IS and annealed NCE is closed and we provide an expression of the estimation error in that limit”.
> > >
> > > On a side note, note that there is a subtle difference between the frameworks of [1] and of our paper:
> > > - in our paper, what is estimated are the log-ratios **directly** [eq 1 our paper]
> > > - in [1], what is estimated are the ratios [eq 15, 1] and **then** the log can be applied
> > >
> > > Both are valid estimators of the log-ratios, but it is not clear that they are the same nor that they have the same MSE. Therefore, showing that the NCE loss is optimal in the MSE sense could be a slightly different result for both cases. It turns out we can show that both estimators do have the same MSE and are happy to include the derivations in the appendix.
> > >
> > > [1] Krause et al. “Algorithms for estimating the partition function of restricted Boltzmann machines.” Artificial Intelligence, 2020.
> > >
> > > - **optimality of the arithmetic path**
> > >
> > > As we understand, the reviewer highlights that the sampling task algorithmically depends on the parameterization of a path, as the parameterization defines which sequence of intermediate distributions will need to be sampled from. In the case of the arithmetic path, the optimal parameterization requires an oracle for calculating Z: this means we cannot sample from the optimal arithmetic path before knowing Z (at least approximately) and this makes the problem potentially computationally more difficult. Is that a fair assessment of the reviewer’s comment?
> > >
> > > We agree and do in fact say in the draft that the optimality of the arithmetic path is highly dependent on having an oracle for Z, for example in the paragraphs preceding Theorems 4 and 5. **We are happy to further clarify this and propose to add the following text:** “Note that the optimality of the arithmetic path requires a re-parameterization in terms of the target normalization: this means that the optimal arithmetic path cannot be sampled from without such an oracle. Such an oracle might be in some instances computationally difficult to implement.”

---

> > > > ### Comment · Reviewer_peRm · 2023-08-14
> > > > **Thanks**
> > > >
> > > > Thanks, i have updated the score.
> > > >
> > > > As a final subnote regarding point 1:
> > > >
> > > > in [1] after eq.16 the estimated quantity C is log Z.

---

### Official Review · Reviewer_q5r2 · 2023-06-27

**Soundness:** 4 excellent
**Presentation:** 3 good
**Contribution:** 3 good
**Rating:** 6
**Confidence:** 3

**Summary:**

This paper investigates the benefits of using bridge distributions to estimate the unknown normalizing constant $Z_1$ of a given density $p_1 = f_1 / Z_1$, with $f_1$ known. There are many ways for estimating a normalizing constant, e.g. importance sampling, bridge sampling, umbrella sampling and noise contrastive estimation. For example, in importance sampling an instrumental density $p_0$ from which it is easy to sample is introduced and the normalizing constant is estimated by weighting the samples from $p_0$ according to the ratio $f_1 / p_0$. Naturally, the accuracy of the estimation depends on how far $p_0$ is from $p_1$ (in Rényi 2-divergence for example or KL), and in most practical cases $p_0$ will be in fact very far from $p_1$. A nice idea then consists in introducing bridge distributions that "simplify" the estimation; we introduce a sequence of distributions $p_k$ (which themselves have intractable normalizing constants) such that the discrepancy between $p_{k+1}$ and $p_k$ is small and use them to estimate the normalizing constant. There are many ways for choosing such a sequence of distributions including geometric path, arithmetic path.
This paper investigates the effect of each design choice: the estimator (they challenge the default choice of IS) and the sequence of bridging distributions. More specifically, their contributions are the following:
- They show that the NCE loss is optimal, whatever the number $K$ of bridging distributions and thus showing that IS is in fact not optimal. This extends the result for $K = 1$.
- The performance of IS is known to suffer from exponential dependence on the dimension and this phenomenon is now sharply quantified. In this paper it is shown that this is also the case for NCE at least for the exponential family, which hints that it should be also the case for more complex densities.
- The hope of using the bridging distributions was to diminish the effect of the dimensionality. It is shown that in the limit of infinite bridge distributions and with geometric paths, the error is no longer exponential but only polynomial.
- Finally, in the presence of an oracle providing the target normalizing constant $Z_1$, they show that the arithmetic path can provide an error independent of the dimension.

**Strengths:**

I find the theoretical contributions of this paper original, interesting and significant. Indeed, I appreciate that the authors have deeply investigated some a priori knowledge that practitioners have, such as the optimality of IS and the effect of the annealed distributions. Furthermore, I very much appreciate that the theoretical results are validated empirically on toy examples.

**Weaknesses:**

The assumptions made in the paper are very strong (perfect sampling for example) and some of the results are only asymptotic in the number of bridge distributions. This makes me doubt about the broad relevance of this paper to the NeurIPS community. Furthermore, although the proposed two step procedure was shown to be numerically optimal, it is not tested numerically in the realistic scenario where one has to resort to MCMC for sampling. I believe that there is no reason why this should not be tested and compared to more traditional methods. Otherwise it is practically useless.



**Questions:**

If I am not mistaken, the estimator $\hat{Z}_1$ in the first line of table 1 is not an estimator of $Z_1$ as $f_0$ is not normalized. Also, in eq. 4 and eq. 5 I believe that there should be $+ \log Z_0$.

**Limitations:**

See weaknesses

---

> ### Author Rebuttal · Authors · 2023-08-08
>
> We thank the reviewer for their valuable feedback, and we are glad the reviewer finds the paper “original, interesting and significant”. We next address comments point-by-point and hope the reviewer will consider raising their score.
>
> **The assumptions made in the paper are very strong [...]  This makes me doubt about the broad relevance of this paper to the NeurIPS community. —**
> While we agree with the reviewer that some of the assumptions we make are quite strong, we would also like to point out that they are fairly standard in the related literature. For example, [2] remark that their “analysis assumes perfect transitions which can be unrealistic in practice because many distributions of interest have separated modes between which mixing is difficult.” The assumptions of “perfect sampling” and of a “continuous path” are commonly made [1, 2, 3, 4] to make the theory tractable to obtain results that can serve as coarse guidelines. This also includes recent works in the NeurIPS and ICML communities [2, 3, 4].  We of course agree that it would be great to further close the gap between theory and practice, and hope that papers like ours will serve as a jumping-off point for such work.
>
> **Typos —** the reviewer is correct about the typo in table 1: f0 should be p0 in order to obtain an estimator of the target normalization. Also, in eqs. 4-5 the plus sign in front of logZ0 vanished. Thank you for picking up on this.
>
> [1] Gelman et al. “Simulating normalizing constants: From importance sampling to bridge sampling to path sampling”. Statistical Science, 1998.
>
> [2] Grosse et al. “Annealing between distributions by averaging moments”. NIPS, 2013.
>
> [3] Brekelmans et al. “All in the Exponential Family: Bregman Duality in Thermodynamic Variational Inference”. ICML, 2020.
>
> [4] Goshtasbpour et al. “Adaptive Annealed Importance Sampling with Constant Rate Progress”. ICML, 2023.

---

> > ### Comment · Reviewer_q5r2 · 2023-08-15
> >
> > I would like to thank the authors for their rebuttal. I believe that my comment on testing the two step procedure using MCMC sampling has not been addressed. I am interested in knowing if the authors have tested this and if they believe that it can bring any improvement wrt to more traditional methods (of course, a negative answer will not influence my score!).

---

> > > ### Author Response · Authors · 2023-08-17
> > > **Answer to reviewer q5r2**
> > >
> > > We thank the reviewer for their question: we have not yet thoroughly tested the two-step procedure using MCMC sampling. We expect the results to vary with the situation: depending on the MCMC sampler and on the target distribution, the mixing times would be different and so would the levels of statistical performance and mismatch to the theory.
> > >
> > > That being said, we inspected the robustness of the two-step method when running experiments using perfect samples, and noticed behavior consistent with what we wrote above. Namely, we noticed that the performance of the two-step procedure strongly depends on the quality of the first step (ie how well Z is pre-estimated using the geometric path).

---

### Official Review · Reviewer_EpxH · 2023-07-04

**Soundness:** 2 fair
**Presentation:** 3 good
**Contribution:** 3 good
**Rating:** 5
**Confidence:** 2

**Summary:**

This paper studies annealing schedules used in noise contrastive estimation (NCE). In particular, it conducts rigorous theoretical analyses of existing annealing schedules used in NCE. The paper's framework is general and yields importance sampling, umbrella sampling, and bridge sampling as special cases. The theory suggests that NCE is more efficient in certain scenario compared to annealed importance sampling (AIS).

**Strengths:**

Disclaimer: I am new to noise contrastive estimation, but familiar with the annealed importance sampling literature.

* The paper provides a rigorous analysis of different annealing schedules used in noise contrastive estimation.
* Previous methods for estimating normalizing constants are obtained as a special case, such as umbrella sampling, ration sampling, and bridge sampling.
* The paper shows that the popularly used geometric annealing schedule is suboptimal.

**Weaknesses:**

Given my lack of expertise in terms of NCE, I will focus my comments on the claims about AIS.

* A major concern with the theoretical comparison against AIS is that the specific instance of AIS considered in this paper is known to be suboptimal. The analysis of AIS nowadays involves the "backward sampling path" formalism (first thoroughly explored in [1], but also tackled in Neal's paper [2]), where the paper's formulation of AIS corresponds to the "suboptimal $L$-kernel" (kernel implicitly used for backward sampling), where the suboptimality is already in the name. There exists attempts to obtain approximately optimal $L$-kernels [3,4].
* This is a minor point, but the fact that the AIS with the suboptimal $L$-kernel scales poorly (which is what is referred to as AIS in the paper) is already known. See p. 335 in [5], p. 11 in [6].
* As such, the claimed scope of the paper is way too broad compared to the actual contribution of the paper. I strongly recommend stating that this is a paper focusing exclusively on noise contrastive estimation in the title, abstract, and introduction.
* Also, I think the paper should be more subtle when comparing performance against importance sampling; with the backward sampling
formalism, you do not need to "assume" perfect sampling; you still have non-asymptotic performance guarantees through [7], and asymptotic guarantees through [1]. So not being able to capture this is theoretically quite critical.





**Questions:**

### Major Comments
* Line 40: As mentioned above, we do have some understanding on how to tune the performance of annealing-based normalizing constant estimation. In fact, although less formal, Neal's paper [2] contains a lengthy discussion on this. Why is this not mentioned?
* Line 189-190: "... there is no definitive theory on the ability of annealing to reduce the statistical cost in a general setup." Again, I believe [2] provides some theory, and with the backward sampling formalism, further analysis is possible as in [3], which is based on the results of [7].
* Line 136-137: This is true for importance sampling, but not necessarily for AIS. In fact, this point in the cited reference [27] is shown with only importance sampling. (Thus, the quotation needs to be corrected.) In AIS, with an optimal $L$-kernel, one can achieve the variance of $i.i.d$ samples regardless of the tails, although whether one can obtain an optimal $L$-kernel is something else.
* Line 65-79: On a similar note, this is a description specific to thermodynamic integration/bridge sampling and is not how newer AIS approaches do things nowadays. In fact, in AIS/SMC, you do not need to choose a path before sampling. You can adaptively determine the sequence based on the estimated quality of samples [6].

### References
Disclaimer: I am not the author or affiliated with the authors of the papers below.
* [1] Del Moral, Pierre, Arnaud Doucet, and Ajay Jasra. "Sequential monte carlo samplers." Journal of the Royal Statistical Society Series B: Statistical Methodology 68.3 (2006): 411-436.
* [2] Neal, Radford M. "Annealed importance sampling." Statistics and computing 11 (2001): 125-139.
* [3] Bernton, Espen, et al. "Schr\" odinger Bridge Samplers." arXiv preprint arXiv:1912.13170 (2019).
* [4] Doucet, Arnaud, et al. "Score-based diffusion meets annealed importance sampling." Advances in Neural Information Processing Systems 35 (2022): 21482-21494.
* [5] Chopin, Nicolas, and Omiros Papaspiliopoulos. An introduction to sequential Monte Carlo. Vol. 4. New York: Springer, 2020.
* [6] Dai, Chenguang, et al. "An invitation to sequential Monte Carlo samplers." Journal of the American Statistical Association 117.539 (2022): 1587-1600.
* [7] Agapiou, Sergios, et al. "Importance sampling: Intrinsic dimension and computational cost." Statistical Science (2017): 405-431.

**Limitations:**

Yes.

---

> ### Author Rebuttal · Authors · 2023-08-08
>
> We thank the reviewer for their valuable feedback and pointing out the relevance of SMC literature to this problem. As we understand, the reviewer broadly makes two points. First, our analysis assumes “perfect sampling” while SMC acknowledges “imperfect sampling” which is more realistic. Second, relevant results from the SMC literature could be better referenced. In the following, we answer these concerns and hope the reviewer will consider raising their score accordingly.
>
> **Our analysis assumes perfect sampling** (the samples are drawn from the path) **while SMC acknowledges imperfect sampling** (the samples are drawn from an MCMC kernel possibly followed by resampling) **which is more realistic** — this is a good point, and we appreciate the chance to clarify. We would like to note that for inexplicit paths of distributions, analyzing the estimation error of AIS/SMC seems unclear and challenging [Eq 38, 1]. In particular, samples from MCMC will typically follow distributions that are not analytically tractable (since they are not easily describable in closed form), thus stronger assumptions seem needed for analysis. We make the assumption of “perfect sampling” and highlight that **this assumption is fairly standard** in the related literature [4, 5, 6, 7].
>
> In fact, **many SMC results referenced by the reviewer also make the “perfect sampling” assumption**. It translates in the SMC literature as:
> - a suboptimal “time reversal” L-kernel (sentence preceding eq 2.5 of [2])
> - the convergence of MCMC steps along the path (assumption 3.1 in [2])
>
> These assumptions are needed to make the estimation error depend explicitly on the path of distributions as in eq 3.2 of [2].
>
>
> **We note that even with the “perfect sampling” assumption, existing results on how the estimation error scales with dimensionality are limited:**
> - Section 3.3 of [2] provides an argument that annealing can reduce the statistical complexity to polynomial in the dimension, but that argument is largely heuristic and relies on restrictive assumptions such as an essentially log-concave path of distributions (assumption 3.4 of [2])
> - Section 4 of [3] discusses the benefits of annealing based on a heuristic calculation
> - Other works mentioned at the end of section 3 of [2], assume independent components of the target distribution which is very restrictive or refer to “relevant discussions” for polynomial dependence in the dimensionality without a definite theory.
>
> **We are happy to include a detailed discussion on these results from SMC literature in the camera-ready version**, to complement lines 40 and 189-190 of our paper, and thank the reviewer for the relevant references they pointed out! To complement lines  65-79 of our paper, we will also discuss versions of AIS where the path is not explicitly defined as in thermodynamic integration, but adaptively defined “on the go” as in [7], as suggested by the reviewer.
>
> The reviewer also mentions some works that do not make the “perfect sampling” assumption. Some of these works target the optimal “backward L-kernel” [8, 9]. We note however that while they report empirical results, they do not theoretically study the estimation error. There are important challenges to a theory that uses Monte Carlo estimates of the optimal “backward L-kernel”: for example, the estimation of the backward kernel (the Stein score in [9]) introduces variance that cannot be ignored and for which the analysis can be quite involved [10].
>
> **Regarding the scope of this paper**, the cost function that we analyze is obtained from seminal AIS literature [4, 5] and we show in our Theorem 1 and this cost function applies just as well to annealed importance sampling, annealed noise-contrastive estimation, and annealed reverse-importance sampling, under the assumption of perfect sampling. We do agree with the reviewer that our analysis of AIS strongly relies on that assumption: we could change the title to something like “Provable benefits of annealing for estimating normalizing constants: IS, NCE and beyond” or “Provable statistical benefits of annealing for estimating normalizing constants”. Hopefully this clarifies that we separate the estimation error of AIS from the uncertainty in the sampling procedure which is “oracled away”.
>
>
> [1] Del Moral. "Sequential monte carlo samplers." Statistical Methodology, 2006.
>
> [2] Dai et al. "An invitation to sequential Monte Carlo samplers." Journal of the American Statistical Association, 2002.
>
> [3] Neal. “Annealed importance sampling”. Statistics and computing, 2001.
>
> [4] Gelman et al. “Simulating normalizing constants: From importance sampling to bridge sampling to path sampling”. Statistical Science, 1998.
>
> [5] Grosse et al. “Annealing between distributions by averaging moments”. NIPS, 2013.
>
> [6] Brekelmans et al. “All in the Exponential Family: Bregman Duality in Thermodynamic Variational Inference”. ICML, 2020.
>
> [7] Goshtasbpour et al. “Adaptive Annealed Importance Sampling with Constant Rate Progress”. ICML, 2023.
>
> [8] Bernton, Espen, et al. "Schrodinger Bridge Samplers." arXiv preprint arXiv:1912.13170 (2019).
>
> [9] Doucet, Arnaud, et al. "Score-based diffusion meets annealed importance sampling." Advances in Neural Information Processing Systems 35 (2022): 21482-21494.
>
> [10] Qin et al. “Fit Like You Sample: Sample-Efficient Generalized Score Matching from Fast Mixing Markov Chains”. Arxiv, 2023.

---

> > ### Comment · Reviewer_EpxH · 2023-08-10
> > **Further Response**
> >
> > I sincerely thank the authors for the detailed response. I strongly believe that our discussions would strengthen and clarify the paper, given the importance and broadness of the field of normalizing constant estimation. I am very happy to update my score, given the constructive engagement of the authors.
> >
> > > We make the assumption of “perfect sampling” and highlight that this assumption is fairly standard in the related literature [4, 5, 6, 7].
> >
> > I agree that the perfect sampling assumption is standard in the literature on bridge sampling and thermodynamic integration. My concern is that AIS/SMC are very different from these. So I believe there is no disagreement here. But I see now (especially from Grosse *et al.*, 2013) that there are subtle differences in what people think is AIS.
> >
> > > In fact, many SMC results referenced by the reviewer also make the “perfect sampling” assumption. It translates in the SMC literature as:
> > >
> > > * a suboptimal “time reversal” L-kernel (sentence preceding eq 2.5 of [2])
> > > * the convergence of MCMC steps along the path (assumption 3.1 in [2])
> >
> > Yes, the optimal L-kernel, or "perfect sampling" assumption, has certainly been mentioned in the SMC literature. However, my point about the L-kernel is that the results of Agapiou *et al.* (2017) allow some theoretical understanding of non-optimal L-kernels as done by Bernton *et al.* (2019). And the CLT results of Del Moral *et al.* (2006) and the likes certainly provide a rigorous theory for arbitrary L-kernels. That is, SMC/(some)AIS papers "assume" perfect sampling in the sense that it ensures the best practical performance, but not for the theory to work.
> >
> > To summarize, my concern is that the fact that the paper oracles away the sampling error reduces its impact/applicability to the SMC/(L-kernel-based)AIS. Given this, the conclusions against (A)IS are too strong. But I do not question the impact on other theoretical frameworks, such as thermodynamic integration (TI) and bridge sampling.
> >
> > > Regarding the scope of this paper, the cost function that we analyze is obtained from seminal AIS literature [4, 5] and we show in our Theorem 1 and this cost function applies just as well to annealed importance sampling, annealed noise-contrastive estimation, and annealed reverse-importance sampling, under the assumption of perfect sampling. We do agree with the reviewer that our analysis of AIS strongly relies on that assumption: we could change the title to something like “Provable benefits of annealing for estimating normalizing constants: IS, NCE and beyond” or “Provable statistical benefits of annealing for estimating normalizing constants”. Hopefully this clarifies that we separate the estimation error of AIS from the uncertainty in the sampling procedure which is “oracled away”.
> >
> > Thank you for the suggestions. Given that the paper operates within the NCE framework, I believe having NCE in the title would certainly be more descriptive.

---

> > > ### Author Response · Authors · 2023-08-12
> > > **Answer to Reviewer EpxH**
> > >
> > > We thank the reviewer for their engagement, as well as updating their score and the discussion points on how AIS is analyzed from an SMC / L-Kernel perspective!

---

### Official Review · Reviewer_6Yhf · 2023-07-06

**Soundness:** 4 excellent
**Presentation:** 4 excellent
**Contribution:** 3 good
**Rating:** 8
**Confidence:** 3

**Summary:**

In this work, the authors make a number of contributions to the area of estimating normalization factors using annealing from some "proposal" $p_0$ to a "target" $p_1$.

The authors start off by nicely extending recent works by Chehab et al. (2023) on the relation between importance sampling (IS) and noise-constrastive estimation (NCE) to the scenario involving annealing, introducing the notion of *annealed Bregman estimators (ABE)* in the proces, computing hte intermediate log-ratios by solving a classificiation task between samples drawn from neighboring densities.

Making in particular use of the ABE defined by the NCE loss, the authors then obtain a number of asymptotic results (in the number of temperatures $K \to \infty$) for the mean-squared error of the estimator of $log Z_1$, i.e. the normalization constant of the target $p_1$.

In Theorem 3, they show that in the case of the commonly used geometric path $p_t = p_0^{1 - \beta} p_1^{\beta}$, both annealed IS and annealed NCE result in an MSE estimate that polynomial in the parameter distance between $p_0$ and $p_1$. This result is established under assumptions of the exponential family for both the proposal and the target, which is a more general setting than seen in previous works. Moreover, the authors also show in Theorem 2, under almost identical assumptions to Theorem 3, that even with the NCE loss instead of IS, the MSE will still scales exponentially in the distance between the parameters of the proposal and target, as is a well-known fact for the IS-based estimator.
Combining these two results, the authors have demonstrated theoretically that *annealing* allows us to move from exponential to polynomial dependence on the parameter distance, effectively bridging the gap between many empirical results from the literature where annealing has been observed to be of much help even in higher dimensions vs. standard importance sampling.

Furthermore, in Theorem 4, 5, and 6, the authors show that the MSE when using the recently introduced arithmetic path $p_t = (1 - \beta) p_0 + \beta p_1$, has exponential, polynomial, and constant dependence on the parameter distance, respectively, under different path-parameterizations. Of course, this path requires the normalization constant in its computation, but the authors use the observation to propose a two-step estimation procedure which they then demonstrate to be useful on a simple toy-problem.

**Strengths:**

I believe this work to be of excellent standard.

The authors are very nicely combining the previous works of Gelman & Meng (1998) and Chehab et al. (2023) to provide both additional theoretical and empirical insights. The theoretical results are, as far as I can tell, both very novel and very relevant. In particular, I find Theorems 2 & 3 to be very pleasing as they give us a clear theoretical motivation for the usage of annealing in higher dimensions, which is a result long sought after. In addition to this, they also bring the work of Chehab et al. (2023) into the annealing setting, and demonstrates its utility both theoretically and empirically (though on toy-problems).

The presentation is of very high quality; the text is well-written, the results are presented in a clear and nice manner, illustrations are used to get the intuition across, and tables are used when appropriate to give the reader a quick overview of results.

As far as I can tell, the authors also do a good job mentioning previous works and the limitations of their own work.

**Weaknesses:**

There are honestly very few weaknesses to point out when it comes to the paper itself; the only weaknesses I can find are related to limitations of the analysis and the resulting proposed practical methods, which I will leave to the "Limitations" section. The one aspect I'd like to raise, though this is slightly related to the aforementioned limitations, is that I would have liked to see slightly more extensive empirical results for the proposed methods. I do agree with the authors in that the empirical results are not the main focus of the work, but still, have a slightly more complex problem would have been nice to see how, for example, the two-step (trig) approach would do in this.

**Questions:**

-   Theorem 6: In Gelman & Meng (1998) in Section 4.3, they make the following comment "we doubt (51) is achievable as it is bounded above by $\pi^2 / n$ even if $p_0$ and $p_1$ are infinitely apart" referring to the continuous-time limit.
    This is then slightly sharper than the $2 \pi^2 / N$ result presented in this work. Can the authors comment on this? Was this just a slightly overly confident statement by Gelman & Meng, i.e. they really meant including the factor of 2?


**Limitations:**

The main limitations of the theoretical analysis are of course the assumptions.

The authors assume that both the proposal and the target are in the exponential family, in addition varying assumptions on the normalization constant of the target as a function of the parameters. This is often not the case in practice, but, as mentioned by the authors, the exponential family some universality properties, which one could argue makes this slightly less of an issue.

As most other works on annealing, this work also assumes perfect sampling from the intermediate distributions, which is usually not the case in practice. But, as mentioned, this is a very common assumption to make. This is also mentioned by the authors.

---

> ### Author Rebuttal · Authors · 2023-08-08
>
> We thank the reviewer for their positive feedback! We also thank the reviewer for spotting the difference between the $2 \pi^2 / N$ in eq. 17 in our paper and the $\pi^2 / N$ from [1] which is correct. This is due to an unfortunate typo: in the supplementary material of our paper, eq 112 defines $\alpha_H = \pi / 4$, hence in eq 117 we should have $\alpha_H^2 = \pi^2 / 16$ which recovers the result of [1].
>
> [1] Gelman et al. “Simulating normalizing constants: From importance sampling to bridge sampling to path sampling”. Statistical Science, 1998.

---

### Author Rebuttal · Authors · 2023-08-08

We thank all four reviewers for their feedback, and for suggesting references which we plan to include in a camera-ready version to give further context to our results. In this general reply, we further clarify the relevance of our results and we address specific concerns in detail in the individual replies.

**Relevance of our results —** annealed importance sampling is a seminal method in statistics and while it is known that the choice of annealing path can greatly impact the estimation error [1], theoretical guarantees on commonly used paths have been elusive. Some prior works have relied on heuristic arguments [2, 3], strong assumptions like gaussianity [3] or essentially log-concavity [2] along the path, or a factorial target [4] to study how the estimation error scales with the dimensionality for a certain choice of path. But in a general setting, no such results exist to our knowledge. **We believe our results are the first to quantify how the geometric path — which has been the standard for more than a decade — actually impacts the estimation error in terms of the dimensionality of the problem**. Our results are also **the first theoretical analysis of the estimation error from an arithmetic path since it was introduced [5]**. We believe these results are relevant to the NeurIPS community, especially as annealing paths are becoming a central component of many recent advances in machine learning [6, 7].





[1] Gelman et al. “Simulating normalizing constants: From importance sampling to bridge sampling to path sampling”. Statistical Science, 1998.

[2] Dai et al. "An invitation to sequential Monte Carlo samplers." Journal of the American Statistical Association, 2002.

[3] Neal. “Annealed importance sampling”. Statistics and computing, 2001.

[4] Beskos et al. “On the stability of sequential Monte Carlo methods in high dimensions’. The Annals of Applied Probability, 2014.

[5] Masrani et al. “q-Paths Generalizing the Geometric Annealing Path using Power Means”. UAI, 2021.

[6] Rhodes et al. “Telescoping Density-Ratio Estimation”. NeurIPS, 2020.

[7] Song et al. “Score-Based Generative Modeling through Stochastic Differential Equations”. ICLR, 2021.

---

### Decision · Program_Chairs · 2023-09-21

**Decision:**

Accept (spotlight)

**Comment:**

The reviewers all agree this is a strong paper that makes a number of significant contributions to our understanding of methods for normalization constant estimation, including providing theoretical support for the good performance of AIS seen in practice.